# WILTing Trees: Interpreting the Distance Between MPNN Embeddings

**Masahiro Negishi   Thomas Gärtner** [1]   **Pascal Welke** [2,1]

## Abstract

We investigate the distance function learned by message passing neural networks (MPNNs) in specific tasks, aiming to capture the *functional* distance between prediction targets that MPNNs implicitly learn. This contrasts with previous work, which links MPNN distances on arbitrary tasks to *structural* distances on graphs that ignore task-specific information. To address this gap, we distill the distance between MPNN embeddings into an interpretable graph distance. Our method uses optimal transport on the Weisfeiler Leman Labeling Tree (WILT), where the edge weights reveal subgraphs that strongly influence the distance between embeddings. This approach generalizes two well-known graph kernels and can be computed in linear time. Through extensive experiments, we demonstrate that MPNNs define the relative position of embeddings by focusing on a small set of subgraphs that are known to be functionally important in the domain.

## 1. Introduction

Message passing graph neural networks (MPNNs) have achieved high predictive performance in various domains (Zhou et al., 2020). To understand these performance gains, researchers have focused on the expressive power of MPNNs (Morris et al., 2019; Xu et al., 2019; Maron et al., 2019). However, the binary nature of expressive power excludes any analysis of the distance between graph embeddings, which is considered to be a key to the predictive power of MPNNs (Liu et al., 2022b; Li & Leskovec, 2022; Morris et al., 2024). Recently, there has been growing interest in the analysis of MPNN (generalization) performance using *structural* distances between graphs (Chuang & Jegelka, 2022; Böker et al., 2024; Franks et al., 2024) that consider graph topology but ignore the target function to be

learned. That line of work has derived generalization bounds under strong assumptions on the margin between classes or on the Lipschitz constants of MPNNs, which often do not hold in practice. In this work, we instead investigate the distance $d_{\mathrm{MPNN}}$ implicitly obtained from a popular MPNN trained on a practical dataset.

Specifically, we ask: *What properties does the distance $d_{MPNN}$ learned by a well-performing MPNN have in practice that can explain its high performance?* While previous studies (Chuang & Jegelka, 2022; Böker et al., 2024) focused on the alignment between $d_{\mathrm{MPNN}}$ and a non-task-tailored structural graph distance $d_{\mathrm{struc}}$, we have found that it is not critical to the predictive performance of MPNNs. Rather, even if an MPNN was trained with classical cross-entropy loss, $d_{\mathrm{MPNN}}$ respects the task-relevant functional distance $d_{\mathrm{func}}$. Furthermore, the alignment between $d_{\mathrm{MPNN}}$ and $d_{\mathrm{func}}$ is highly correlated with the predictive performance of the MPNN. Then, we move to our second question: *How do MPNNs learn such a metric structure?* Since $d_{\mathrm{MPNN}}$ is essentially a distance between multisets of Weisfeiler Leman (WL) subgraphs, we distill $d_{\mathrm{MPNN}}$ into a more interpretable distance between these multisets. Our distance $d_{\mathrm{WILT}}$ is an optimal transport distance on a *weighted Weisfeiler Leman Labeling Tree* (WILT), which is a trainable generalization of the graph distances of existing high-performance kernels (Kriege et al., 2016; Togninalli et al., 2019). It allows us to identify WL subgraphs whose presence or absence significantly affects the relative position of graphs in the MPNN embedding space. In addition, $d_{\mathrm{WILT}}$ is efficiently computable as the ground metric is a path length on a tree, and is at least as expressive as $d_{\mathrm{MPNN}}$ in terms of binary expressive power. We show experimentally that the WILTing tree distances fit MPNN distances well. Examination of the resulting edge parameters on a WILT after distillation shows that only a small number of WL subgraphs determine $d_{\mathrm{MPNN}}$. In a qualitative experiment, the subgraphs that strongly influence $d_{\mathrm{MPNN}}$ are those that are known to be functionally important by domain knowledge. In short, our contributions are as follows:

- We show that MPNN distances after training are aligned with the task-relevant functional distance of the graphs and that this is key to the high predictive performance of MPNNs.

---

[1]TU Wien, Vienna, Austria [2]Lancaster University Leipzig, Leipzig, Germany. Correspondence to: Masahiro Negishi <m.negishi25@imperial.ac.uk>.

*Proceedings of the $42^{nd}$ International Conference on Machine Learning*, Vancouver, Canada. PMLR 267, 2025. Copyright 2025 by the author(s).

- We propose a trainable graph distance on a weighted Weisfeiler-Lehman Labeling Tree (WILT) that generalizes Weisfeiler Leman-based distances and is efficiently computable.

- WILT allows a straightforward definition of *relevant* subgraphs. Thus, distilling an MPNN into a WILT enables us to identify subgraphs that strongly influence the distance between MPNN embeddings, allowing an interpretation of the MPNN embedding space.

## 2. Related Work

Recently, it has become increasingly recognized that the geometry of the MPNN embedding space, not just its binary expressiveness, is crucial to its performance (Li & Leskovec, 2022; Morris et al., 2024). For instance, graph contrastive learning methods implicitly assume that good metric structure in the embedding space leads to high performance (Liu et al., 2022b). Chuang & Jegelka (2022); Böker et al. (2024); Franks et al. (2024) analyzed MPNN's (generalization) performance using *structural* distances between graphs, which consider graph topology but ignore the learning task at hand. Despite their theoretically sound analyses of generalization bounds, they made strong assumptions about the Lipschitz constants of MPNNs (Chuang & Jegelka, 2022; Böker et al., 2024) or the margin between classes (Franks et al., 2024). In addition, Böker et al. (2024) dealt only with dense graphs and required the consideration of all MPNNs with some Lipschitz constant. Our study also focuses on the geometry of the embedding space, but we empirically investigate practical MPNNs trained on real, sparse graphs, mainly using task-dependent *functional* distances without limiting Lipschitz constants or the margin between classes.

This study is also related to GNN interpretability (Liu et al., 2022a; Yuan et al., 2022). Higher interpretability of well-performing models may lead to a new understanding of scientific phenomena when applied to scientific domains such as chemistry or biology. In addition, it may be a requirement in real-world problems where safety or privacy is critical. Most of the existing interpretation methods are instance-level, identifying input features in a given input graph that are important for its prediction. However, instance-level methods cannot explain the global behavior of GNNs. Recently, some studies have proposed a way to understand the global behavior of GNNs by distilling them into highly interpretable models. The resulting model can be a GNN with higher interpretability (Müller et al., 2024), or a logical formula (Azzolin et al., 2023; Köhler & Heindorf, 2024; Pluska et al., 2024). Our study also distills an MPNN into a highly interpretable WILT for global-level understanding. This allows to interpret the metric structure of the MPNN embedding space, while previous studies focused on generating explanations for each label class. In addition,

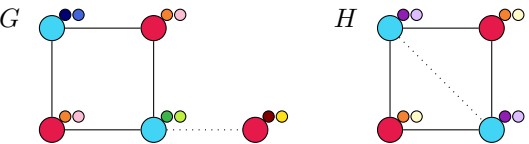

Figure 1: Example of two iterations of the Weisfeiler Leman algorithm. ● and ● are colors corresponding to initial node attributes, while — and ⋯ represent edges with different attributes. Node colors in iterations one and two are shown in the small circles next to the nodes. For example, ● = HASH(●, ⦃(●, −), (●, −)⦄) and ● = HASH(●, ⦃(●, −), (●, −), (●, ⋯)⦄), where HASH is a perfect hash function.

our method can be applied to graph regression tasks, while previous studies are restricted to classification problems.

## 3. Preliminaries

We define a graph as a tuple $G = (V, E, l_{\text{node}}, l_{\text{edge}})$, where $V$ and $E$ are the set of nodes and edges. Each node and each edge have an attribute defined by $l_{\text{node}} : V \rightarrow \Sigma_{\text{node}}$ and $l_{\text{edge}} : E \rightarrow \Sigma_{\text{edge}}$, where $\Sigma_{\text{node}}$ and $\Sigma_{\text{edge}}$ are finite sets. We restrict them to finite sets because our method is based on the Weisfeiler Leman test described below, which is discrete in nature. We denote the set of all graphs up to isomorphism as $\mathcal{G}$ and the set of neighbors of node $v$ as $\mathcal{N}(v)$. We only consider undirected graphs, but the extension to directed graphs is easy by employing an appropriate version of the Weisfeiler Leman test.

**Message Passing Algorithms** (Gilmer et al., 2017) include popular GNNs such as Graph Convolutional Networks (GCN, Kipf & Welling, 2017), and Graph Isomorphism Networks (GIN, Xu et al., 2019). At each iteration, a message passing algorithm updates the embeddings of all nodes by aggregating the embeddings of themselves and their neighbors in the previous iteration. After $L$ iterations, the node embeddings are aggregated into the graph embedding $h_G$:

$$a_v^{(l)} = \text{AGG}^{(l)}\left(\{\!\!\{(h_u^{(l-1)}, l_{\text{edge}}(e_{vu})) \mid u \in \mathcal{N}(v)\}\!\!\}\right),$$
$$h_v^{(l)} = \text{UPD}^{(l)}\left(h_v^{(l-1)}, a_v^{(l)}\right),$$
$$h_G = \text{READ}\left(\{\!\!\{h_v^{(L)} \mid v \in V\}\!\!\}\right).$$

Here, $\{\!\!\{\cdot\}\!\!\}$ denotes a multiset, and $0 < l \leq L$ with $h_v^{(0)} = l_{\text{node}}(v)$. $h_v^{(l)} \in \mathbb{R}^d$ and $h_G \in \mathbb{R}^{d'}$ are the embedding of node $v$ after the $l$-th layer and the graph embedding, respectively. $\text{AGG}^{(l)}$, $\text{UPD}^{(l)}$, and READ are functions. *Message Passing Neural Networks* (MPNNs) implement $\text{UPD}^{(l)}$ and $\text{AGG}^{(l)}$ using multilayer perceptrons (MLPs). Sum and mean pooling are popular for READ.

**The Weisfeiler Leman (WL) Algorithm** is a message passing algorithm, where $\text{UPD}^{(l)}$ is an injective function. $\text{AGG}^{(l)}$ and READ are the identity function on multisets. A node embedding of the WL algorithm is called *color*. We use $c_v^{(l)}$ instead of $h_v^{(l)}$ to refer to it. Figure 1 shows the progress of the WL algorithm on two graphs: $G$ and $H$ start with the same colors, but no longer share colors after two iterations, i.e., $\{\!\{c_v^{(2)} \mid v \in V_G\}\!\} \cap \{\!\{c_v^{(2)} \mid v \in V_H\}\!\} = \emptyset$.

**Message Passing Pseudometrics** The WL algorithm cannot distinguish some nonisomorphic graphs (Cai et al., 1992) and all MPNNs are bounded by its expressiveness (Xu et al., 2019). Hence, any MPNN yields a pseudometric on the set of pairwise nonisomorphic graphs $\mathcal{G}$.

**Definition 3.1** (Graph Pseudometric). A graph pseudometric space $(\mathcal{G}, d)$ is given by a non-negative real valued function $d : \mathcal{G} \times \mathcal{G} \to \mathbb{R}_{\geq 0}$ that satisfies for all $F, G, H \in \mathcal{G}$:

$$d(G, G) = 0 \qquad \textbf{(Identity)}$$
$$d(G, H) = d(H, G) \qquad \textbf{(Symmetry)}$$
$$d(G, F) \leq d(G, H) + d(H, F) \quad \textbf{(Triangle inequality)}$$

Given an MPNN, we obtain a pseudometric space $(\mathcal{G}, d_{\text{MPNN}})$ by setting $d_{\text{MPNN}}(G, H) := d(h_G, h_H)$, where $d : \mathbb{R}^{d'} \times \mathbb{R}^{d'} \to \mathbb{R}$ is a (pseudo)metric and $h_G$ and $h_H$ are graph embeddings. Note that $(\mathcal{G}, d_{\text{MPNN}})$ is not a metric space since there are nonisomorphic graphs $G, H$ with identical representations and hence $d_{\text{MPNN}}(G, H) = 0$. For the rest of this paper, we will use $d_{\text{MPNN}}(G, H) = ||h_G - h_H||_2$, but other distances between embeddings can also be used. Note that $d_{\text{MPNN}}$ depends not only on the input graphs but also on the task on which the MPNN is trained. For example, $d_{\text{MPNN}}$ of an MPNN trained to predict the toxicity of molecules will be different from $d_{\text{MPNN}}$ of another MPNN trained to predict the solubility of the same molecules.

**Structural Pseudometrics** To date, many different graph kernels have been proposed (see Kriege et al., 2020). Each positive semidefinite graph kernel $k : \mathcal{G} \times \mathcal{G} \to \mathbb{R}$ corresponds to a pseudometric between graphs. Please see Appendix A.1 for the definitions of structural pseudometrics from previous studies that are used in this article. We will refer to these pseudometrics as *structural* pseudometrics and write $d_{\text{struc}}$, as they only consider the structural and node/edge attribute information of graphs, without being trained using the target label information.

**Functional Pseudometrics** To formally define the functional distance between graphs, we introduce another pseudometric on $\mathcal{G}$ that is based on the target labels of the graphs.

**Definition 3.2** (Functional Pseudometric). Let $y_G$ be the target label of graph $G$ in a given task. In classification, $y_G$ is a categorical class, while $y_G$ is a numerical value in regression. We assume the space for $y_G$ is bounded. Then, the

functional pseudometric space $(\mathcal{G}, d_{\text{func}})$ is obtained from $d_{\text{func}} : \mathcal{G} \times \mathcal{G} \to [0, 1]$ defined as:

$$d_{\text{func}}(G, H) := \begin{cases} \mathbb{1}_{y_G \neq y_H} & \text{(classification)} \\ \dfrac{|y_G - y_H|}{\sup\limits_{I \in \mathcal{G}} y_I - \inf\limits_{I \in \mathcal{G}} y_I} & \text{(regression)}, \end{cases}$$

where $\mathbb{1}_{y_G \neq y_H}$ is the indicator function that returns 1 if $y_G \neq y_H$, otherwise 0.

See Appendix A.2 for a proof that $(\mathcal{G}, d_{\text{func}})$ is a pseudometric space. If the sup/inf of $y_G$ in $\mathcal{G}$ are unknown, they can be approximated by the max/min in a training dataset.

**The Expressive Power** of a message passing algorithm is defined based on its ability to distinguish non-isomorphic graphs. Formally, a message passing graph embedding function $f$ is said to be at least as expressive as another one $g$ if the following holds:

$$\forall G, H \in \mathcal{G} : f(G) = f(H) \implies g(G) = g(H),$$

where $\mathcal{G}$ is the set of all pairwise non-isomorphic graphs. We extend the above to pseudometrics on graphs. Specifically, a graph pseudometric $d$ is said to be at least as expressive as $d'$ ($d \geq d'$) iff

$$\forall G, H \in \mathcal{G} : d(G, H) = 0 \implies d'(G, H) = 0.$$

$d$ and $d'$ are equally expressive ($d \cong d'$) iff $d \geq d'$ and $d' \geq d$. Furthermore, $d$ is said to be more expressive than $d'$ ($d > d'$) iff $d \geq d'$ and there exists $G, H \in \mathcal{G}$ s.t. $d(G, H) \neq 0 \wedge d'(G, H) = 0$.

## 4. Is the MPNN Embedding Distance Critical to Performance?

Our first question is what properties $d_{\text{MPNN}}$ of well performing MPNNs have in practice that can explain their high performance. This section investigates whether the alignment between $d_{\text{MPNN}}$ and the *task-relevant* pseudometric $d_{\text{func}}$ is such a property. Specifically, we address the questions below:

**Q1.1** Does training an MPNN increase the alignment between $d_{\text{MPNN}}$ and the task-relevant $d_{\text{func}}$?

**Q1.2** Does a strong alignment between $d_{\text{MPNN}}$ and $d_{\text{func}}$ indicate high performance of the MPNN?

Note that the alignment between $d_{\text{MPNN}}$ and *task-irrelevant* structural graph pseudometrics $d_{\text{struc}}$ has been considered a key to MPNN performance in previous studies (Chuang & Jegelka, 2022; Böker et al., 2024; Franks et al., 2024). However, we found that this property is not consistently improved by training and does not correlate with performance. (See Appendix D for detailed analyses).

To answer **Q1.1** and **Q1.2**, we will first define a measure of the alignment between $d_{\text{MPNN}}$ and $d_{\text{func}}$. Note that it is inappropriate to adopt a typical min/max of $\frac{d_{\text{func}}(G,H)}{d_{\text{MPNN}}(G,H)}$ to measure the alignment. This is because $d_{\text{func}}$ is a binary function for classification tasks, and expecting the exact match of the two distances is unreasonable. Thus, we define our evaluation criterion as follows.

**Definition 4.1** (Evaluation Criterion for Alignment Between $d_{\text{MPNN}}$ and $d_{\text{func}}$)**.** Let $\mathcal{D}$ be a graph dataset, $k(\geq 1)$ be an integer hyperparameter, and $\mathcal{N}_k(G) \subset \mathcal{D} \setminus \{G\}$ be a set of $k$ graphs that are closest to $G$ under $d_{\text{MPNN}}$. Let

$$A_k(G) := \frac{1}{k} \sum_{H \in \mathcal{N}_k(G)} d_{\text{func}}(G, H),$$

$$B_k(G) := \frac{1}{|\mathcal{D}| - k - 1} \sum_{H \in \mathcal{D} \setminus (\mathcal{N}_k(G) \cup \{G\})} d_{\text{func}}(G, H).$$

Then, $d_{\text{MPNN}}$ is *aligned* with $d_{\text{func}}$ if

$$\text{ALI}_k(d_{\text{MPNN}}, d_{\text{func}}) := \frac{1}{|\mathcal{D}|} \sum_{G \in \mathcal{D}} \left[ -A_k(G) + B_k(G) \right]$$

is positive. In addition, the larger $\text{ALI}_k$ is, the more we say $d_{\text{MPNN}}$ is aligned with $d_{\text{func}}$.

Here, $A_k(G)$ and $B_k(G)$ are the average functional distances between $G$ and its neighbors and non-neighbors, respectively. If $A_k(G) < B_k(G)$, then $G$ and closeby graphs in MPNN space have a lower average functional distance than $G$ and far away graphs in MPNN space.

We show the distribution of $\text{ALI}_k(d_{\text{MPNN}}, d_{\text{func}})$ for 48 different MPNNs on different datasets and varying $k$ in Figure 2. Each model was trained with a standard loss function (cross entropy loss for classification and RMSE for regression). We did not explicitly optimize $\text{ALI}_k$. We also include the results for untrained MPNNs to see the effect of training. We can see that there is little overlap between the distributions of the untrained and trained MPNNs. This means that $\text{ALI}_k$ consistently improves through training, implying a positive answer to **Q1.1**. Next, we compute Spearman's rank correlation coefficient (SRC) between $\text{ALI}_k(d_{\text{MPNN}}, d_{\text{func}})$ of trained MPNNs and their predictive performance. We use accuracy and RMSE between the ground truth target and predicted values to measure classification and regression performance, respectively. Table 1 shows that SRC for Mutagenicity and ENZYMES is always positive, indicating that the higher the $\text{ALI}_k$, the higher the accuracy. Similarly, the higher the $\text{ALI}_k$, the lower the RMSE for Lipophilicity. The correlations are consistent across training and test sets. These results suggest that the degree of alignment between $d_{\text{MPNN}}$ and $d_{\text{func}}$ is a crucial factor contributing to the high performance of MPNNs, answering **Q1.2** positively. See Appendix C for more details and additional results on non-molecular datasets.

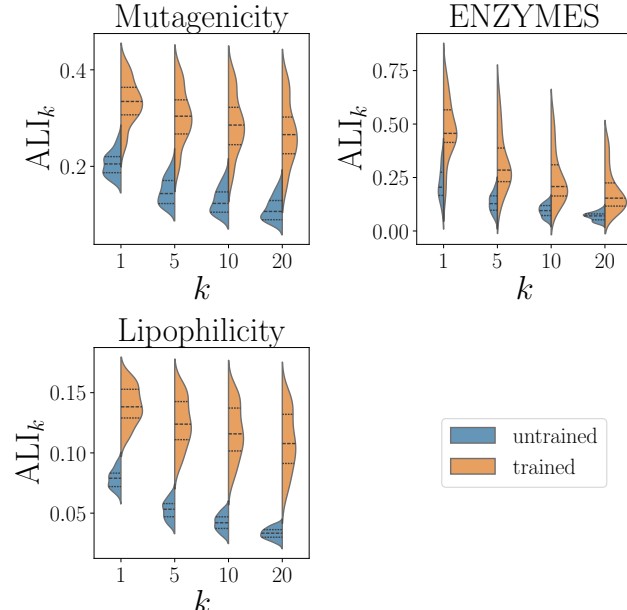

Figure 2: The distribution of $\text{ALI}_k(d_{\text{MPNN}}, d_{\text{func}})$ under different $k$ and datasets.

Table 1: Correlation (SRC) between $\text{ALI}_k(d_{\text{MPNN}}, d_{\text{func}})$ and the performance on $\mathcal{D}_{\text{train}}$ and $\mathcal{D}_{\text{test}}$ under different $k$. We use accuracy for Mutagenicity and ENZYMES, and RMSE for Lipophilicity to measure performance.

| k | Mutagenicity | | ENZYMES | | Lipophilicity | |
|---|---|---|---|---|---|---|
| | train | test | train | test | train | test |
| 1 | 0.66 | 0.70 | 0.89 | 0.49 | -0.65 | -0.57 |
| 5 | 0.65 | 0.69 | 0.87 | 0.50 | -0.63 | -0.56 |
| 10 | 0.63 | 0.68 | 0.87 | 0.47 | -0.62 | -0.56 |
| 20 | 0.61 | 0.67 | 0.85 | 0.46 | -0.60 | -0.55 |

## 5. WILTing Pseudometrics

Section 4 confirms that MPNNs are implicitly trained so that $d_{\text{MPNN}}$ aligns with $d_{\text{func}}$, which turns out to be crucial for MPNN's performance. Then, our second research question is: how do MPNNs learn $d_{\text{MPNN}}$ that respects $d_{\text{func}}$? To this end, we note that MPNN embeddings are aggregations of WL color embeddings, i.e., there exists a function $f$ with $h_G = f(\{\!\{c_v \mid v \in V_G\}\!\})$. Hence, $d_{\text{MPNN}}$ defines a pseudometric $d_f$ between multisets of WL colors

$$\begin{aligned} d_{\text{MPNN}}(G, H) &:= \|h_G - h_H\|_2 \\ &= \|f(\{\!\{c_v^{(L)} \mid v \in V_G\}\!\}) - f(\{\!\{c_v^{(L)} \mid v \in V_H\}\!\})\|_2 \\ &=: d_f(\{\!\{c_v^{(L)} \mid v \in V_G\}\!\}, \{\!\{c_v^{(L)} \mid v \in V_H\}\!\}). \end{aligned}$$

Therefore, we aim to understand $d_{\text{MPNN}}$ by distilling it into our more interpretable and equally expressive pseudometric $d_{\text{WILT}}$ on the same multisets. $d_{\text{WILT}}$ is an optimal transport

distance on the weighted Weisfeiler Leman Labeling Tree (WILT) and generalizes two existing pseudometrics of high-performing graph kernels (Kriege et al., 2016; Togninalli et al., 2019). After distillation, the edge parameters of $d_{\text{WILT}}$ allow us to identify WL colors whose presence or absence significantly affects the relative position of graphs in the MPNN embedding space. In addition, $d_{\text{WILT}}$ is efficient to compute as the ground metric is a path length on the tree.

### 5.1. Weisfeiler Leman Labeling Tree (WILT)

The Weisfeiler Leman Labeling Tree (WILT) $T_{\mathcal{D}}$ is a rooted weighted tree built from the set of colors obtained by the WL test on a graph dataset $\mathcal{D} \subseteq \mathcal{G}$. Given $\mathcal{D}$, we define $V(T_{\mathcal{D}})$ as the *set* of colors that appear on any node during the WL test plus the root node $r$, that is, $V(T_{\mathcal{D}}) = \{c_v^{(l)} \mid v \in V_G, G \in \mathcal{D}, l \in [L]\} \cup \{r\}$. Colors $x, y \in V(T_{\mathcal{D}}) \setminus \{r\}$ are adjacent if and only if there exists a node $v$ in some graph in $\mathcal{D}$ and an iteration $l$ with $x = c_v^{(l)}$ and $y = c_v^{(l-1)}$. $r$ is connected to all $x = c_v^{(0)}$. Due to the injectivity of the AGG and UPD functions in the WL algorithm, it follows that $T_{\mathcal{D}}$ is a tree. Figure 3 (top) shows the WILT built from the graphs $G$ and $H$ in Figure 1. See Appendix B for a detailed algorithm to build a WILT from $\mathcal{D}$.

We consider edge weights $w : E(T_{\mathcal{D}}) \to \mathbb{R}_{\geq 0}$ on WILT. We only allow non-negative weights so that the WILTing distance in Definition 5.1 will be non-negative. Given a WILT $T_{\mathcal{D}}$ with weights $w$, the shortest path length $d_{\text{path}}(x, y; w) := \sum_{e \in \text{Path}(x,y)} w(e)$ is the sum of edge weights of the unique shortest path $\text{Path}(x, y)$ between $x$ and $y$. Note that $d_{\text{path}}$ is a pseudometric on $V(T_{\mathcal{D}})$, i.e., the set of WL colors in $\mathcal{D}$. Intuitively, $d_{\text{path}}(x, y; w)$ is large if $\text{Path}(x, y)$ is long, but $w$ allows us to tune this pseudometric according to the needs of the learning task.

### 5.2. The WILTing Distance

A WILT $T_{\mathcal{D}}$ with edge weights $w$ yields a pseudometric $d_{\text{WILT}}$ on the graph set $\mathcal{D}$. This section shows two equivalent characterizations of $d_{\text{WILT}}$ as an optimal transport distance and as a weighted Manhattan distance. The latter allows us to define the importance of specific WL colors and to compute our proposed pseudometric efficiently. For simplicity, we define $d_{\text{WILT}}$ for graphs with the same number of nodes. In the next section, we will discuss the extension to graphs with different numbers of nodes. For two distributions with identical mass on the same pseudometric space, optimal transport distances such as the Wasserstein distance (Villani, 2009) measure the minimum effort of shifting probability mass from one distribution to the other. Each unit of shifted mass is weighted by the distance it is shifted. We define our pseudometric $d_{\text{WILT}}(G, H; w)$ as the optimal transport between $V_G$ and $V_H$, where the ground pseudometric is the shortest path metric on the WILT $T_{\mathcal{D}}$.

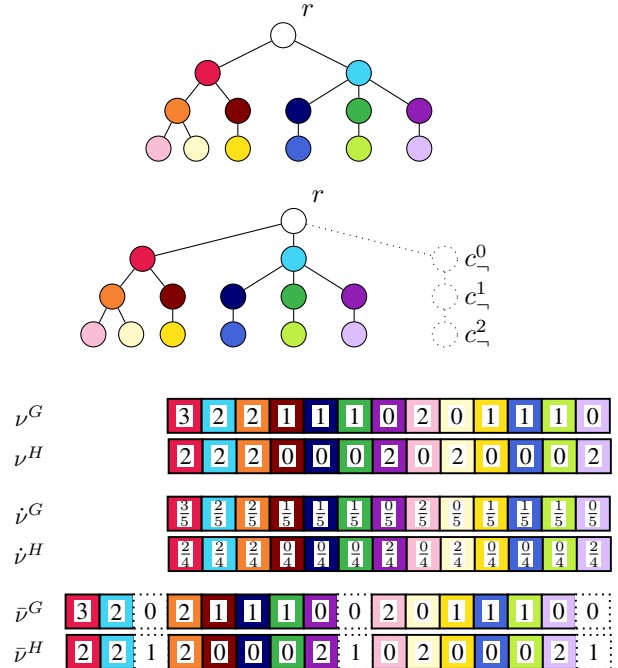

Figure 3: (top): The Weisfeiler Leman Labelling Tree (WILT) built from $\mathcal{D} = \{G, H\}$ from Figure 1. (middle): The WILT built from $\mathcal{D} = \{G, H\}$ with dummy nodes. (bottom): The WILT embeddings $\nu, \dot{\nu}$ with size normalization, and $\bar{\nu}$ with dummy node normalization.

**Definition 5.1** (WILTing Distance). Let $G, H \in \mathcal{D}$ be graphs with $|V_G| = |V_H|$. Then

$$d_{\text{WILT}}(G, H; w) := \min_{P \in \Gamma} \sum_{v_i \in V_G} \sum_{u_j \in V_H} P_{i,j} d_{\text{path}}(c_{v_i}^{(L)}, c_{u_j}^{(L)})$$

where $\Gamma := \{P \in \mathbb{R}^{|V_G| \times |V_H|} \mid P_{i,j} \geq 0, P\mathbf{1} = \mathbf{1}, P^T\mathbf{1} = \mathbf{1}\}$.

Note that $d_{\text{WILT}}$ is not a metric but a pseudometric on the set of pairwise nonisomorphic graphs $\mathcal{G}$. This is because there are nonisomorphic graphs $G$ and $H$ whose colors are the same after $L$ iterations, i.e., $\{\!\{c_v^{(L)} \mid v \in V_G\}\!\} = \{\!\{c_v^{(L)} \mid v \in V_H\}\!\}$.

Generic algorithms to compute Wasserstein distances require cubic runtime. In our case, however, there exists a *linear* time algorithm to compute $d_{\text{WILT}}$ as shown below, since the ground pseudometric $d_{\text{path}}$ is the shortest path metric on a tree (Le et al., 2019).

**Definition 5.2** (WILT Embedding). The WILT embedding of a graph $G \in \mathcal{D}$ is a vector, where each dimension counts how many times a corresponding WL color appears during the WL test on $G$, i.e., $\nu_c^G := |\{v \in V_G \mid \exists l \in [L]\, c_v^{(l)} = c\}|$ for $c \in V(T_{\mathcal{D}}) \setminus \{r\}$. (see the bottom of Figure 3).

**Proposition 5.3** (Equivalent Definition of WILTing Distance). *$d_{WILT}$ in Definition 5.1 is equivalent to:*

$$d_{WILT}(G, H; w) = \sum_{c \in V(T_{\mathcal{D}}) \setminus \{r\}} w\left(e_{\{c, p(c)\}}\right) \left|\nu_c^G - \nu_c^H\right|,$$

*where $e_{\{c, p(c)\}}$ is the edge connecting $c$ and its parent $p(c)$ in $T_{\mathcal{D}}$.*

This equivalence allows efficient computation of $d_{\text{WILT}}$ given the WILT embeddings of graphs, which can be computed by the WL algorithm in $O(L|E_G|)$ time, where $L$ is the number of WL iterations. Using sparse vectors for $\nu^G$ and $\nu^H$, $d_{\text{WILT}}(G, H)$ can be computed in $O(|V_G| + |V_H|)$.

### 5.3. Normalization and Special Cases

The definition of $d_{\text{WILT}}(G, H)$ as an optimal transport distance requires $|V_G| = |V_H|$. However, $|V_G|$ and $|V_H|$ are usually different, so we propose two solutions. Interestingly, the two modified WILTing distances generalize two pseudometrics corresponding to well-known graph kernels.

**Size Normalization** Straightforwardly, we can restrict the mass of each node to $\frac{1}{|V_G|}$ when calculating the Wasserstein distance in Definition 5.1. In other words, we replace $\Gamma$ with $\dot{\Gamma} := \{P \in \mathbb{R}^{|V_G| \times |V_H|} \mid P_{i,j} \geq 0, P\mathbf{1} = \frac{1}{|V_G|}\mathbf{1}, P^T\mathbf{1} = \frac{1}{|V_H|}\mathbf{1}\}$. Similarly, $\nu^G$ in Proposition 5.3 is changed to $\dot{\nu}^G := \frac{\nu^G}{|V_G|}$. The resulting pseudometric $\dot{d}_{\text{WILT}}$ effectively ignores differences in the number of nodes of $G$ and $H$, generally assigning fractions of colors in $G$ to colors in $H$. In Figure 3 (bottom), we show $\dot{\nu}$ of $G$ and $H$ from Figure 1. $\dot{d}_{\text{WILT}}(G, H)$ is calculated as:

$$\dot{d}_{WILT}(G, H) = w(e_{\{\bullet, \circ\}}) \left|\frac{3}{5} - \frac{2}{4}\right| + w(e_{\{\circ, \circ\}}) \left|\frac{2}{5} - \frac{2}{4}\right|$$
$$+ \ldots + w(e_{\{\circ, \bullet\}}) \left|\frac{0}{5} - \frac{2}{4}\right|.$$

An interesting property of $\dot{d}_{\text{WILT}}$ is that it generalizes the pseudometric corresponding to the Wasserstein Weisfeiler Leman graph kernel (Togninalli et al., 2019): when $w \equiv \frac{1}{2(L+1)}$, $\dot{d}_{\text{WILT}}$ matches their pseudometric. See Appendix A.3 for technical details.

**Dummy Node Normalization** We can also add isolated nodes with a special attribute, called dummy nodes, to graphs so that all the graphs have the same number of nodes. The WILT will be built in the same way as described in Section 5.1 after dummy nodes are added to all graphs in $\mathcal{D}$. The resulting WILT has new colors $c_{\neg}^0, c_{\neg}^1, \ldots, c_{\neg}^L$ that arise from the WL iteration on the isolated dummy nodes (Figure 3 middle). The WILT embedding will be slightly changed to

$$\bar{\nu}_c^G := \begin{cases} N - |V_G| & \text{if } c \in \{c_{\neg}^0, c_{\neg}^1, \ldots, c_{\neg}^L\} \\ \nu_c^G & \text{otherwise} \end{cases},$$

---

**Algorithm 1** Optimizing edge weights of WILT

**Input:** graph dataset $\mathcal{D}$, an MPNN $f$ with $L$ message passing layers trained on $\mathcal{D}$, and WILT $T_{\mathcal{D}}$ built from the results of $L$-iteration WL test on $\mathcal{D}$
**Parameter:** batch size, number of epochs $E$, and learning rate $lr$
**Output:** learned edge weights $w$ of WILT $T_{\mathcal{D}}$
$n_c \leftarrow |E(T_{\mathcal{D}})|$
$w \leftarrow \mathbb{1} \in \mathbb{R}^{n_c}$
optimizer $\leftarrow$ Adam(params=$w$, lr=$lr$)
**for** $e = 1$ **to** $E$ **do**
   **for** batch $B$ in $\mathcal{D}^2$ **do**
      $l \leftarrow \frac{1}{|B|} \sum\limits_{(G,H) \in B} \left(d_{\text{WILT}}(G, H) - d_{\text{MPNN}}(G, H)\right)^2$
      $l$.backward()
      optimizer.step()
      /* Ensuring that edge weights $w$ are non-negative */
      $w \leftarrow \max(w, 0)$
**return** $w$

---

where $N = \max_{G \in \mathcal{D}} |V_G|$ (Figure 3 bottom). Then, the resulting pseudometric $\bar{d}_{\text{WILT}}(G, H)$ for the graphs in Figure 1 is:

$$\bar{d}_{WILT}(G, H) = w(e_{\{\bullet, \circ\}})|3 - 2| + w(e_{\{\circ, \circ\}})|2 - 2|$$
$$+ \ldots + w(e_{\{\circ, \circ\}})|0 - 1|.$$

Similar to size normalization, $\bar{d}_{\text{WILT}}$ includes the pseudometric of Weisfeiler Leman optimal assignment kernel (Kriege et al., 2016) as a special case. When $w \equiv \frac{1}{2}$, $\bar{d}_{\text{WILT}}$ is equivalent to their pseudometric. See Appendix A.3 for details.

### 5.4. Edge Weight Learning and Identification of Important WL Colors

Now, we have a graph pseudometric on WILT defined for any pairs of graphs in $\mathcal{D}$. Next, we show how to optimize the edge weights $w$. Proposition 5.3 allows us to learn the edge weights $w$, given training data. Specifically, given a target pseudometric $d_{\text{target}}$ we adapt $d_{\text{WILT}}$ by minimizing

$$\mathcal{L}(w) := \sum_{(G, H) \in \mathcal{D}^2} \left(d_{\text{WILT}}(G, H; w) - d_{\text{target}}(G, H)\right)^2,$$

with respect to $w$. Note that $d_{\text{WILT}}$ can refer to both $\dot{d}_{\text{WILT}}$ and $\bar{d}_{\text{WILT}}$. In this work, we focus on $d_{\text{target}} = d_{\text{MPNN}}$. That is, we train $d_{\text{WILT}}$ to mimic the distances between the graph embeddings of a given MPNN, as shown in Algorithm 1. Once we have trained $w$ by minimizing $\mathcal{L}$, we can gain insight into $d_{\text{MPNN}}$ via $d_{\text{WILT}}$. WL colors with large edge weights are those whose presence or absence in a graph significantly affects $d_{\text{MPNN}}$ between the graph and other

graphs. Specifically, we can derive the following reasoning.

Large difference between $G$ and $H$ in the number
or ratio of WL colors $c$ with a large $w(e_{\{c,p(c)\}})$

$\implies$ Large $d_{\text{WILT}}(G, H)$   ($\because$ Proposition 5.3)

$\implies$ Large $d_{\text{MPNN}}(G, H)$   ($\because$ $d_{\text{WILT}}$ approximates $d_{\text{MPNN}}$)

### 5.5. Expressiveness of Pseudometrics on WILT

Here, we discuss which of the two normalizations is preferred for a given MPNN based on the expressive power. Below are the relationships between the expressiveness of $d_{\text{MPNN}}$ and $d_{\text{WILT}}$.

**Theorem 5.4** (Expressive Power of the Pseudometrics on WILT). *Let $d_{MPNN}^{mean}$ and $d_{MPNN}^{sum}$ be $d_{MPNN}$ of MPNNs with mean and sum graph poolings, respectively. We also define a pseudometric based on the L-iteration WL test:*

$$d_{WL}(G, H) := \mathbb{1}_{\{\!\{c_v^{(L)}|v \in V_G\}\!\} \neq \{\!\{c_v^{(L)}|v \in V_H\}\!\}}.$$

*Then, for WILT with positive edge weights, the following relationships hold between the expressive power of each pseudometric.*

$$\dot{d}_{WILT} < \bar{d}_{WILT} \cong d_{WL},$$
$$d_{MPNN}^{mean} \leq \dot{d}_{WILT}(< \bar{d}_{WILT}),$$
$$d_{MPNN}^{sum} \leq \bar{d}_{WILT}, \ d_{MPNN}^{sum} \lesseqgtr \dot{d}_{WILT}.$$

*Proof.* See Appendix A.4.                        □

Since $\bar{d}_{\text{WILT}}$ is more expressive than $\dot{d}_{\text{WILT}}$, one might think that $\bar{d}_{\text{WILT}}$ is always preferable to approximating $d_{\text{MPNN}}$. However, $\dot{d}_{\text{WILT}}$ is expected to be better at approximating $d_{\text{MPNN}}^{\text{mean}}$, since it provides a tighter bound. Intuitively, this follows from the fact that mean pooling and the size normalization are essentially the same procedure: they both ignore the number of nodes. In contrast, $\bar{d}_{\text{WILT}}$ is expected to work well on $d_{\text{MPNN}}^{\text{sum}}$, which retains the information about the number of nodes and thus cannot be bounded by $\dot{d}_{\text{WILT}}$. We will experimentally confirm these analyses in Section 6. Note that Theorem 5.4 considers only the binary expressiveness of pseudometrics. Regarding the size of the family of functions that each pseudometric can represent, $d_{\text{MPNN}}$ is expected to be superior to $d_{\text{WILT}}$, because $d_{\text{WILT}}$ is restricted to an optimal transport on the tree for faster computation and better interpretability. Still, in Section 6, we empirically show that $d_{\text{WILT}}$ can approximate $d_{\text{MPNN}}$ well.

## 6. Experiments

In this section, we confirm that our proposed $d_{\text{WILT}}$ can successfully approximate $d_{\text{MPNN}}$. Then, we show that the

Table 2: The mean±std of RMSE($d_{\text{MPNN}}, d$) [$\times 10^{-2}$] over five different seeds. Each row corresponds to a GCN with a given graph pooling method, trained on a given dataset.

|  | $d_{\text{WWL}}$ | $d_{\text{WLOA}}$ | $\dot{d}_{\text{WILT}}$ | $\bar{d}_{\text{WILT}}$ |
|---|---|---|---|---|
| Mutagenicity |  |  |  |  |
| mean | 9.25±0.87 | 18.74±3.36 | 1.74±0.52 | 3.34±1.01 |
| sum | 12.25±0.54 | 5.98±1.60 | 1.22±0.31 | 0.82±0.17 |
| ENZYMES |  |  |  |  |
| mean | 12.18±0.23 | 16.79±2.33 | 2.71±0.38 | 4.64±0.67 |
| sum | 11.28±0.65 | 6.83±0.41 | 9.15±0.47 | 1.43±0.10 |
| Lipophilicity |  |  |  |  |
| mean | 10.92±0.42 | 13.97±0.97 | 3.11±0.54 | 6.35±1.22 |
| sum | 10.83±0.73 | 10.00±1.34 | 2.50±0.67 | 2.64±0.74 |

distribution of learned edge weights of WILT is skewed towards 0, and a large part of them can be removed with L1 regularization. Finally, we investigate the WL colors that influence $d_{\text{MPNN}}$ most. Due to space limitations, we report results only for a selection of MPNNs and datasets. Code is available online, and experimental settings and additional results are in Appendix E.

We trained 3-layer GCNs with mean or sum pooling on the three datasets with five different seeds. We then distilled each into two WILTs, one with size normalization and one with dummy node normalization. To evaluate how well a distance $d$ approximates $d_{\text{MPNN}}$, we used a variant of RMSE:

$$\text{RMSE}(d_{\text{MPNN}}, d)$$
$$:= \sqrt{\min_{\alpha \in \mathbb{R}} \frac{1}{|\mathcal{D}|^2} \sum_{(G,H) \in \mathcal{D}^2} \left( \hat{d}_{\text{MPNN}}(G, H) - \alpha \cdot \hat{d}(G, H) \right)^2},$$

where $\hat{d}_{\text{WILT}}$ and $\hat{d}$ means they are normalized to $[0, 1]$. Intuitively, the closer the RMSE is to zero, the better the alignment is, and zero RMSE means perfect alignment. We do not use the correlation coefficient because it can be one even if $d_{\text{MPNN}}$ is not a constant multiple of $d$: it allows a non-zero intercept. Note that the minimization over $\alpha$ can be solved analytically. In practice, we compute the RMSE using only 1000 pairs from $\mathcal{D}^2$ for speed. Table 2 shows the RMSE between $d_{\text{MPNN}}$ and $\dot{d}_{\text{WILT}}$ or $\bar{d}_{\text{WILT}}$. We also include results for $d_{\text{WWL}}$ and $d_{\text{WLOA}}$, which are special cases of $\dot{d}_{\text{WILT}}$ and $\bar{d}_{\text{WILT}}$ with fixed edge weights, respectively. It is obvious that $d_{\text{WILT}}$ aligns with $d_{\text{MPNN}}$ much better than $d_{\text{WWL}}$ and $d_{\text{WLOA}}$. Interestingly, $\dot{d}_{\text{WILT}}$ approximates $d_{\text{MPNN}}$(mean) better, while $\bar{d}_{\text{WILT}}$ approximates $d_{\text{MPNN}}$(sum) better, except for $d_{\text{MPNN}}$(sum) trained on Lipophilicity, where their performance is close. This observation is consistent with the theoretical analysis in Section 5.5.

Next, we look into the distribution of the learned edge weights of WILT. Figure 4 shows the histogram of the edge weights of the WILT with dummy node normalization after

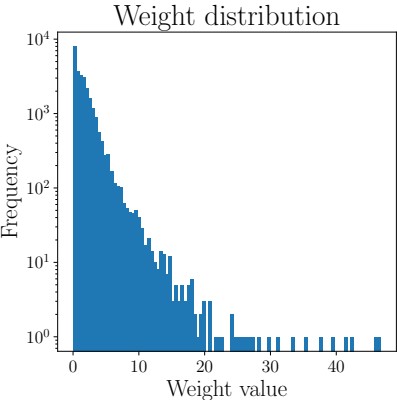

Figure 4: The distribution of the edge weights of WILT after distillation.

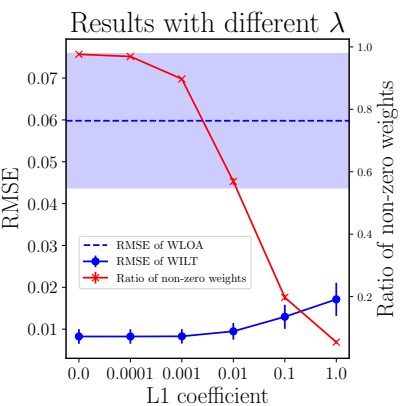

Figure 5: The RMSE and the ratio of non-zero edge weights after distillation under different L1 coefficients (mean and std over five different seeds).

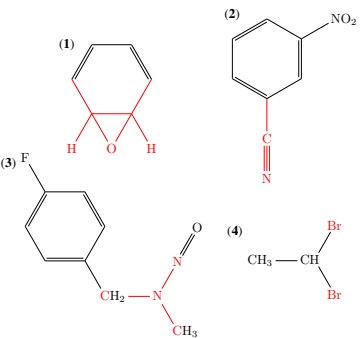

Figure 6: Example graphs with high-lighted significant subgraphs corresponding to colors with the largest weights.

distillation from a 3-layer GCN with sum pooling trained on Mutagenicity. The distribution is heavily skewed towards zero. This plot, together with Proposition 5.3, suggests that the relative position of MPNN graph embeddings is determined based on only a small subset of WL colors. To further verify this idea, we added an L1 regularization term to the objective function $\mathcal{L}$ and minimized it so that $w(e_{\{c,p(c)\}})$ would be set to zero for some colors. Figure 5 shows the RMSE between $d_{\text{MPNN}}$ and the resulting $\bar{d}_{\text{WILT}}$, as well as the ratio of non-zero edge weights, under different L1 coefficient $\lambda$. As expected, the larger $\lambda$ is, the more edge weights are set to zero and the larger the RMSE. However, it is worth noting that $\bar{d}_{\text{WILT}}$ is much better aligned with $d_{\text{MPNN}}$ than $d_{\text{WLOA}}$ even when trained with $\lambda = 1.0$ and about 95% of the edge weights are zero. This good approximation with only 5% non-zero edges implies that MPNNs rely on only a few important WL colors to define $d_{\text{MPNN}}$.

Finally, we show the subgraphs corresponding to the colors with the largest weights, thus influencing $d_{\text{MPNN}}$ the most. Again, we only show results for the 3-layer GCN with sum pooling trained on the Mutagenicity dataset. To avoid identifying colors that are too rare, we only consider colors that appear in at least 1% of the entire graphs. Figure 6 shows example graphs with subgraphs corresponding to colors with the four largest weights. The identified subgraphs in (1) and (4) are known to be characteristic of mutagenic molecules (Kazius et al., 2005). In fact, (1) and (4) are classified as "epoxide" and "aliphatic halide" based on the highlighted subgraphs. Given that only a tiny fraction of the entire WL colors correspond to the subgraphs reported in (Kazius et al., 2005), this result suggests that MPNNs learn the relative position of graph embeddings based on WL colors that are also known to be functionally important by domain knowledge.

## 7. Conclusions

We analyzed the metric properties of the embedding space of MPNNs. We found that the alignment with the functional pseudometric improves during training and is a key to high predictive performance. In contrast, the alignment with the structural psudometrics does not improve and is not correlated with performance. To understand how MPNNs learn and reflect the functional distance between graphs, we propose a theoretically sound and efficiently computable new pseudometric on graphs using WILT. By examining the edge weights of the distilled WILT, we found that only a tiny fraction of the entire WL colors influence $d_{\text{MPNN}}$. The identified colors correspond to subgraphs that are known to be functionally important from domain knowledge.

One limitation of our study is that we only distilled two GNN architectures (GCN and GIN) with fixed hyperparameters to WILT. Thus, it remains to be seen how different architectures and hyperparameters affect the edge weights of WILT. We also limited our analysis to the final embeddings of MPNNs, but in principle, WILT can be trained to approximate the MPNN embedding distance at internal layers. It would be interesting to investigate the embedding distances at different layers, and how they relate to the performance. We expect results similar to Liu et al. (2024), which showed that consistency between distances at different iterations is a key to high performance. While we investigated MPNNs specifically, there is a hierarchy of more and more expressive GNNs that are bounded in expressiveness by corresponding WL test variants. In this paper, we have defined WILT on the hierarchy of 1-WL labels. Still, it is straightforward to extend the proposed WILT metric to color hierarchies obtained from higher-order WL variants (Morris et al., 2023; Geerts & Reutter, 2022) or extended message passing schemes (Frasca et al., 2022; Graziani et al., 2024). While beyond

the scope of this work, higher-order WILTing trees may prove useful in interpreting a range of GNNs. However, as the number of trainable WILT weights scales with the number of colors, the practical relevance of higher-order WILTs remains an open question. It is also worth exploring extending WILT to the analysis of GNNs for node classification on large graphs such as social networks or citation networks. Kothapalli et al. (2023) showed that GNNs are trained such that their *node* embeddings respect the functional alignment, but their analysis was limited to unrealistic graphs generated from a stochastic block model. We believe that the node embedding space of GNNs can also be distilled to WILT in a similar way, i.e., by tuning the weights to approximate the node embedding distance with a path distance on WILT. Using WILT for a purpose other than understanding GNNs is also interesting. For example, by training WILT's edge parameters from scratch, we might be able to build a high-performance, interpretable graph learning method.

## Acknowledgements

PW acknowledges TGs ability to devise great paper titles involving puns on some authors' last names. This work was supported by the Vienna Science and Technology Fund (WWTF) and the City of Vienna project StruDL (10.47379/ ICT22059) and by the Austrian Science Fund (FWF) project NanOX-ML (6728). We express our gratitude to the Japan Student Services Organization for financially supporting the first author's five-month research stay in Vienna. Finally, we thank the reviewers for their insightful comments.

## Impact Statement

This paper advances our understanding of what and how competitive MPNNs learn. We hope that this will contribute to safe and robust machine learning. We are not aware of any immediate negative consequences of our work.

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

# A. Theoretical Analysis

In this section, we present the proofs of our theoretical results in the main paper. We formally define related structural pseudometrics and show that $d_{\text{WILT}}$ generalizes them.

## A.1. Structural Pseudometrics

Here we introduce the definitions of the graph edit distance ($d_{\text{GED}}$, Sanfeliu & Fu, 1983), Weisfeiler Leman optimal assignment distance ($d_{\text{WLOA}}$, Kriege et al., 2016), and Wasserstein Weisfeiler Leman graph distance ($d_{\text{WWL}}$, Togninalli et al., 2019). For the definition of tree mover's distance, please refer to the original paper (Chuang & Jegelka, 2022).

**Definition A.1** (Graph Edit Distance (Sanfeliu & Fu, 1983)). Let $\mathcal{E}$ be the set of graph edit operations, and $c : \mathcal{E} \to \mathbb{R}_{\geq 0}$ be a function that assigns a cost to each operation. Then, the *graph edit distance* (GED) between $G$ and $H$ is defined as the minimum cost of a sequence of edit operations that transform $G$ into $H$. Formally,

$$d_{\text{GED}}(G, H) := \min_{s \in S(G,H)} \sum_{e \in s} c(e),$$

where $S(G, H)$ is a set of sequences of graph edit operations that transform $G$ into $H$.

In the experiments in Appendix D, $\mathcal{E}$ consists of insertion and deletion of single nodes and single edges, as well as substitution of single node or edge attributes. We set the cost of each operation to 1, i.e., $c(e) \equiv 1$. Next, we move on to the Weisfeiler Leman optimal assignment (WLOA) kernel.

**Definition A.2** (Weisfeiler Leman Optimal Assignment Kernel (Kriege et al., 2016)). Consider $G = (V_G, E_G)$ and $H = (V_H, E_H)$. Let $V'_G$ and $V'_H$ be the extended node sets resulting from adding special nodes $z$ to $G$ or $H$ so that $G$ and $H$ have the same number of nodes. Let the base kernel $k$ be defined as:

$$k(v, u) := \begin{cases} \sum_{l=0}^{L} \mathbb{1}_{c_v^{(l)} = c_u^{(l)}} & (v \neq z \wedge u \neq z) \\ 0 & (v = z \vee u = z), \end{cases}$$

where $c_v^{(l)}$ and $c_u^{(l)}$ represent the colors of vertices $v$ and $u$ at iteration $l$ of the WL algorithm (see Section 3). Then, the Weisfeiler Leman optimal assignment (WLOA) kernel is defined as:

$$k_{\text{WLOA}}(G, H) := \max_{B \in \mathcal{B}(V'_G, V'_H)} \sum_{(v_G, u_H) \in B} k(v_G, u_H),$$

where $\mathcal{B}(V'_G, V'_H)$ denotes the set of all possible bijections between $V'_G$ and $V'_H$.

Kriege et al. (2016) proved that $k_{\text{WLOA}}$ is a positive semidefinite kernel function. While they focused only on the kernel, a corresponding graph pseudometric can be defined in the following way:

**Definition A.3** (Weisfeiler Leman Optimal Assignment (WLOA) Distance). The function $d_{\text{WLOA}}$ below is a pseudometric on the set of pairwise nonisomorphic graphs $\mathcal{G}$:

$$d_{\text{WLOA}}(G, H) := (L + 1) \cdot \max(|V_G|, |V_H|) - k_{\text{WLOA}}(G, H)$$

*Proof.* Theorem A.8 shows that $d_{\text{WLOA}}$ defined as above is a special case of $\bar{d}_{\text{WILT}}$. Since $\bar{d}_{\text{WILT}}$ is a pseudometric on the set of pairwise nonisomorphic graphs $\mathcal{G}$, so is $d_{\text{WLOA}}$. $\square$

We will show later that the above WLOA distance is a special case of our WILT distance with dummy node normalization (Theorem A.8). Togninalli et al. (2019) proposed another graph pseudometric based on the WL algorithm, called Wasserstein Weisfeiler Leman graph distance.

**Definition A.4** (Wasserstein Weisfeiler Leman (WWL) Distance (Togninalli et al., 2019)). Let $d_{\text{ham}}(v, u)$ be the hamming distance between $\left[c_v^{(0)}, c_v^{(1)}, \ldots c_v^{(L)}\right]$ and $\left[c_u^{(0)}, c_u^{(1)}, \ldots c_u^{(L)}\right]$, where $c_v^{(l)}$ is the color of node $v$ at iteration $l$ of the WL algorithm (see Section 3). Specifically,

$$d_{\text{ham}}(v, u) := \frac{1}{L + 1} \sum_{l=0}^{L} \mathbb{1}_{c_v^{(l)} \neq c_u^{(l)}}.$$

Then the WWL distance is defined as

$$d_{\text{WWL}}(G, H) := \min_{P \in \Gamma_{\text{WWL}}} \sum_{v_i \in V_G} \sum_{u_j \in V_H} P_{i,j} d_{\text{ham}}(v_i, u_j),$$

where $\Gamma_{\text{WWL}} := \{P \in \mathbb{R}_{\geq 0}^{|V_G| \times |V_H|} \mid P\mathbf{1} = \frac{1}{|V_G|}\mathbf{1}, P^T\mathbf{1} = \frac{1}{|V_H|}\mathbf{1}\}$ is a set of valid transports between two uniform discrete distributions.

Togninalli et al. (2019) have shown that $d_{\text{WWL}}$ is a pseudometric. In addition, they proposed a corresponding kernel $k_{\text{WWL}}(G, H) := e^{-\lambda d_{\text{WWL}}(G,H)}$, and showed that it is positive semidefinite. We will prove later in Theorem A.7 that our WILT distance with size normalization includes this WWL distance as a special case.

## A.2. Functional Pseudometrics

Here, we show that $d_{\text{func}}$ is a pseudometric.

**Definition 3.2** (Functional Pseudometric). Let $y_G$ be the target label of graph $G$ in a given task. In classification, $y_G$ is a categorical class, while $y_G$ is a numerical value in regression. We assume the space for $y_G$ is bounded. Then, the functional pseudometric space $(\mathcal{G}, d_{\text{func}})$ is obtained from $d_{\text{func}} : \mathcal{G} \times \mathcal{G} \to [0, 1]$ defined as:

$$d_{\text{func}}(G, H) := \begin{cases} \mathbb{1}_{y_G \neq y_H} & \text{(classification)} \\ \frac{|y_G - y_H|}{\sup\limits_{I \in \mathcal{G}} y_I - \inf\limits_{I \in \mathcal{G}} y_I} & \text{(regression)}, \end{cases}$$

where $\mathbb{1}_{y_G \neq y_H}$ is the indicator function that returns 1 if $y_G \neq y_H$, otherwise 0.

*Proof.* We start with the classification case.

$$\begin{aligned} d_{\text{func}}(G, G) &= \mathbb{1}_{y_G \neq y_G} \\ &= 0 \\ d_{\text{func}}(G, H) &= \mathbb{1}_{y_G \neq y_H} \\ &= \mathbb{1}_{y_H \neq y_G} \\ &= d_{\text{func}}(H, G) \\ d_{\text{func}}(G, F) &= \mathbb{1}_{y_G \neq y_F} \\ &\leq \mathbb{1}_{y_G \neq y_H} + \mathbb{1}_{y_H \neq y_F} \\ &= d_{\text{func}}(G, H) + d_{\text{func}}(H, F) \end{aligned}$$

For the regression case we can assume that $\sup\limits_{I \in \mathcal{G}} y_I \neq \inf\limits_{I \in \mathcal{G}} y_I$ as the regression problem is trivial otherwise. Then we can

show:

$$d_{\text{func}}(G, G) = \frac{|y_G - y_G|}{\sup\limits_{I \in \mathcal{G}} y_I - \inf\limits_{I \in \mathcal{G}} y_I}$$

$$= 0$$

$$d_{\text{func}}(G, H) = \frac{|y_G - y_H|}{\sup\limits_{I \in \mathcal{G}} y_I - \inf\limits_{I \in \mathcal{G}} y_I}$$

$$= \frac{|y_H - y_G|}{\sup\limits_{I \in \mathcal{G}} y_I - \inf\limits_{I \in \mathcal{G}} y_I}$$

$$= d_{\text{func}}(H, G)$$

$$d_{\text{func}}(G, F) = \frac{|y_G - y_F|}{\sup\limits_{I \in \mathcal{G}} y_I - \inf\limits_{I \in \mathcal{G}} y_I}$$

$$\leq \frac{|y_G - y_H|}{\sup\limits_{I \in \mathcal{G}} y_I - \inf\limits_{I \in \mathcal{G}} y_I} + \frac{|y_H - y_F|}{\sup\limits_{I \in \mathcal{G}} y_I - \inf\limits_{I \in \mathcal{G}} y_I}$$

$$= d_{\text{func}}(G, H) + d_{\text{func}}(H, F)$$

In both cases, identity, symmetry, and triangle inequality are satisfied. □

If the supremum (resp. infimum) of $y_G$ in $\mathcal{G}$ are unknown, they can be approximated by the maximum (resp. minimum) distance in a training dataset $\mathcal{D}$, and we can similarly prove that $d_{\text{func}}$ is a pseudometric.

### A.3. Normalized WILTing Distances and Relationship to Existing Distances

We present the formal definitions of the size normalization and dummy node normalization. We then show that $\dot{d}_{\text{WILT}}$ with size normalization generalizes the WWL distance $d_{\text{WWL}}$ and that $\bar{d}_{\text{WILT}}$ with dummy node normalization generalizes the WLOA distance $d_{\text{WLOA}}$.

**Definition A.5** (WILTing Distance with Size Normalization)**.** We define the WILTing distance with size normalization as:

$$\dot{d}_{\text{WILT}}(G, H; w) := \min_{P \in \dot{\Gamma}} \sum_{v_i \in V_G} \sum_{u_j \in V_H} P_{i,j} d_{\text{path}}(c_{v_i}^{(L)}, c_{u_j}^{(L)}),$$

where $\dot{\Gamma} := \{P \in \mathbb{R}^{|V_G| \times |V_H|} \mid P_{i,j} \geq 0, P\mathbf{1} = \frac{1}{|V_G|}\mathbf{1}, P^T\mathbf{1} = \frac{1}{|V_H|}\mathbf{1}\}$. It is equivalent to:

$$\dot{d}_{\text{WILT}}(G, H; w) = \sum_{c \in V(T_{\mathcal{D}}) \setminus \{r\}} w(e_{\{c, p(c)\}}) \left| \dot{\nu}_c^G - \dot{\nu}_c^H \right|,$$

where $\dot{\nu}^G := \frac{1}{|V_G|} \nu^G$.

The only difference between Definition 5.1 and Definition A.5 is the mass assigned to each node. The equivalence between the two definitions of $\dot{d}_{\text{WILT}}$ is a straightforward consequence of (Le et al., 2019). The other normalization is defined as follows.

**Definition A.6** (WILTing Distance with Dummy Node Normalization)**.** Let $\bar{V}_G$ be an extension of $V_G$ with additional $N - |V_G|$ isolated dummy nodes with a special attribute, where $N := \max_{G \in \mathcal{D}} |V_G|$. Let $\bar{T}_{\mathcal{D}}$ be WILT built from the extended graphs $\{(\bar{V}_G, E_G)\}_{G \in \mathcal{D}}$. Note that $\bar{T}_{\mathcal{D}}$ is just a slight modification of $T_{\mathcal{D}}$ (see Figure 3). We define the WILTing distance with dummy node normalization as:

$$\bar{d}_{\text{WILT}}(G, H; w) := \min_{P \in \bar{\Gamma}} \sum_{v_i \in \bar{V}_G} \sum_{u_j \in \bar{V}_H} P_{i,j} d_{\text{path}}(\bar{c}_{v_i}^{(L)}, \bar{c}_{u_j}^{(L)}),$$

where $\bar{\Gamma} := \{P \in \mathbb{R}^{|\bar{V}_G| \times |\bar{V}_H|} \mid P_{i,j} \geq 0, P\mathbf{1} = \mathbf{1}, P^T\mathbf{1} = \mathbf{1}\}$, and $\bar{c}_v^{(L)}$ is the color of node $v$ on $\bar{T}_\mathcal{D}$ after $L$ iterations. An equivalent definition is:

$$\bar{d}_{\text{WILT}}(G, H; w) = \sum_{\bar{c} \in V(\bar{T}_\mathcal{D}) \backslash \{r\}} w(e_{\{\bar{c}, p(\bar{c})\}}) \left| \bar{\nu}_{\bar{c}}^G - \bar{\nu}_{\bar{c}}^H \right|,$$

where $\bar{\nu}^G$ is the WILT embedding of $G$ using $\bar{T}_\mathcal{D}$.

Intuitively speaking, we add dummy nodes to all the graphs so that they have the same number of nodes[1], and compute the WILTing distance in exactly the same way as shown in Section 5.2.

Next, we show that $\dot{d}_{\text{WILT}}$ includes the Wasserstein Weisfeiler Leman distance and $\bar{d}_{\text{WILT}}$ includes the Weisfeiler Leman optimal assignment distance as a special case, respectively.

**Theorem A.7** ($d_{\text{WWL}}$ as A Special Case of $\dot{d}_{\text{WILT}}$). *The WWL distance in Definition A.4 is equal to the WILTing distance with size normalization with all WILT edge weights set to $\frac{1}{2(L+1)}$. Specifically,*

$$d_{\text{WWL}}(G, H) = \dot{d}_{\text{WILT}}\left(G, H; w \equiv \frac{1}{2(L+1)}\right).$$

*Proof.*

$$d_{\text{WWL}}(G, H) := \min_{P \in \Gamma_{\text{WWL}}} \sum_{v_i \in V_G} \sum_{u_j \in V_H} P_{i,j} d_{\text{ham}}(v_i, u_j)$$

$$= \min_{P \in \Gamma_{\text{WWL}}} \sum_{v_i \in V_G} \sum_{u_j \in V_H} P_{i,j} \frac{1}{L+1} \sum_{l=0}^{L} \mathbb{1}_{c_v^l \neq c_u^l}$$

$$= \min_{P \in \Gamma_{\text{WWL}}} \sum_{v_i \in V_G} \sum_{u_j \in V_H} P_{i,j} d_{\text{path}}\left(c_{v_i}^{(L)}, c_{u_j}^{(L)}; w \equiv \frac{1}{2(L+1)}\right)$$

$$= \min_{P \in \dot{\Gamma}} \sum_{v_i \in V_G} \sum_{u_j \in V_H} P_{i,j} d_{\text{path}}\left(c_{v_i}^{(L)}, c_{u_j}^{(L)}; w \equiv \frac{1}{2(L+1)}\right)$$

$$= \dot{d}_{\text{WILT}}\left(G, H; w \equiv \frac{1}{2(L+1)}\right).$$

$\square$

**Theorem A.8** ($d_{\text{WLOA}}$ as A Special Case of $\bar{d}_{\text{WILT}}$). *The WLOA distance in Definition A.3 is equal to the WILTing distance with dummy node normalization with all WILT edge weights set to $\frac{1}{2}$. Specifically,*

$$d_{\text{WLOA}}(G, H) = \bar{d}_{\text{WILT}}\left(G, H; w \equiv \frac{1}{2}\right).$$

*Proof.* First, $d_{\text{WLOA}}(G, H)$ can be transformed as follows.

$$d_{\text{WLOA}}(G, H) := (L+1) \cdot \max(|V_G|, |V_H|) - k_{\text{WLOA}}(G, H)$$

$$= (L+1) \cdot \max(|V_G|, |V_H|) - \max_{B \in \mathcal{B}(V_G', V_H')} \sum_{(v_G, u_H) \in B} k(v_G, u_H)$$

$$= \min_{B \in \mathcal{B}(V_G', V_H')} \sum_{(v_G, u_H) \in B} (L + 1 - k(v_G, u_H))$$

Next, we introduce an equivalent definition of $k(v, u)$. In Definition A.2, the WL algorithm is applied only on $V_G$ and $V_H$, not on special nodes. Assume w.l.o.g. that $|V(G)| \leq |V(H)|$, i.e., $V(G)$ is extended with $|V(H)| - |V(G)|$

---

[1]In fact, $\bar{d}_{\text{WILT}}$ remains a pseudometric even on $\mathcal{D} = \mathcal{G}$, as it can be defined without explicit use of $N$. To this end, note that $\lim_{N \to \infty} |\bar{\nu}_{c_\lrcorner^i}^G - \bar{\nu}_{c_\lrcorner^i}^H| = |V(G) - V(H)|$ for any dummy node color $c_\lrcorner^i$.

dummy nodes. By treating the special nodes in $V'_G$ as dummy nodes, we can define WL colors for the special nodes $z$: $(c_z^{(0)}, c_z^{(1)}, \ldots, c_z^{(L)}) = (c_{\lrcorner}^0, c_{\lrcorner}^1, \ldots, c_{\lrcorner}^L)$. Then, as only $V'_G$ contains special nodes, $k(v, u)$ can be simplified to:

$$k(v, u) = \sum_{l=0}^{L} \mathbb{1}_{\bar{c}_v^{(l)} = \bar{c}_u^{(l)}},$$

where $\bar{c}_v^{(l)}$ is the color of node $v$ on the WILT $\bar{T}_{\mathcal{D}}$ with dummy node normalization after $l$ iterations. Then, $L + 1 - k(v, u)$ is equivalent to $d_{\text{path}}(\bar{c}_v^{(L)}, \bar{c}_u^{(L)}; w \equiv \frac{1}{2})$:

$$
\begin{aligned}
L + 1 - k(v, u) &= L + 1 - \sum_{l=0}^{L} \mathbb{1}_{\bar{c}_v^{(l)} = \bar{c}_u^{(l)}} \\
&= \sum_{l=0}^{L} \mathbb{1}_{\bar{c}_v^{(l)} \neq \bar{c}_u^{(l)}} \\
&= d_{\text{path}}\left( \bar{c}_v^{(L)}, \bar{c}_u^{(L)}; w \equiv \frac{1}{2} \right)
\end{aligned}
$$

Therefore, $d_{\text{WLOA}}$ is a special case of $\bar{d}_{\text{WILT}}$:

$$
\begin{aligned}
d_{\text{WLOA}}(G, H) &= \min_{B \in \mathcal{B}(V'_G, V'_H)} \sum_{(v_G, u_H) \in B} (L + 1 - k(v_G, u_H)) \\
&= \min_{B \in \mathcal{B}(V'_G, V'_H)} \sum_{(v_G, u_H) \in B} d_{\text{path}}\left( \bar{c}_{v_G}^{(L)}, \bar{c}_{u_H}^{(L)}; w \equiv \frac{1}{2} \right) \\
&\stackrel{\star}{=} \min_{P \in \bar{\Gamma}} \sum_{v_i \in \bar{V}_G} \sum_{u_j \in \bar{V}_H} P_{i,j} d_{\text{path}}\left( \bar{c}_{v_i}^{(L)}, \bar{c}_{u_j}^{(L)}; w \equiv \frac{1}{2} \right) \\
&= \bar{d}_{\text{WILT}}\left( G, H; w \equiv \frac{1}{2} \right)
\end{aligned}
$$

Note that $\star$ holds since adding the same number of dummy nodes to both $G$ and $H$ does not change the left side, and the optimal transport on WILT always delivers a mass on a node to only one node. □

### A.4. Expressiveness of Graph Pseudometrics

We now discuss in detail the expressiveness of graph pseudometrics, which was summarized in Section 5.5. We split Theorem 5.4 in Section 5.5 into three theorems below, and prove each one separately. The discussion below provides a possible explanation for some results in Section 6 and Appendices D and E. First, we introduce a pseudometric defined by the WL test:

$$d_{\text{WL}}(G, H) \coloneqq \mathbb{1}_{\{\!\{c_v^{(L)} | v \in V_G\}\!\} \neq \{\!\{c_v^{(L)} | v \in V_H\}\!\}},$$

where $L$ is the number of WL iterations. In other words, $d_{\text{WL}}(G, H) = 1$ if the $L$-iteration WL test can distinguish $G$ and $H$, otherwise 0. With this definition, we start with the comparison of $d_{\text{WILT}}$ and $d_{\text{WL}}$ for a better understanding of $d_{\text{WILT}}$.

**Theorem A.9** (Expressiveness of the WILTing Distance). *Suppose $\dot{d}_{\text{WILT}}$ and $\bar{d}_{\text{WILT}}$ are pseudometrics defined with WILT with some edge weight functions. We assume that all edge weights are positive for $\bar{d}_{\text{WILT}}$. Then,*

$$\dot{d}_{\text{WILT}} < \bar{d}_{\text{WILT}} \cong d_{\text{WL}}.$$

*Proof.* We first show $\dot{d}_{\text{WILT}} \leq \bar{d}_{\text{WILT}}$.

$$\bar{d}_{\text{WILT}}(G, H) = 0 \implies \bar{\nu}^G = \bar{\nu}^H \quad \wedge \quad |V_G| = |V_H|$$
$$\implies \forall \text{ leaf color } c: \quad |\{v \in V_G \mid c_v^{(L)} = c\}| = |\{v \in V_H \mid c_v^{(L)} = c\}| \quad \wedge \quad |V_G| = |V_H|$$
$$\implies \forall \text{ leaf color } c: \quad \frac{|\{v \in V_G \mid c_v^{(L)} = c\}|}{|V_G|} = \frac{|\{v \in V_H \mid c_v^{(L)} = c\}|}{|V_H|}$$
$$\implies \dot{\nu}^G = \dot{\nu}^H$$
$$\implies \dot{d}_{\text{WILT}}(G, H) = 0.$$

Note that leaf color $c$ means that $c$ is a leaf of the WILT. The first implication follows from the fact that dummy node normalization implies that only graphs with identical numbers of nodes can have a distance of zero if the weights are positive. To see that $\bar{d}_{\text{WILT}}(G, H)$ is more expressive than $\dot{d}_{\text{WILT}}(G, H)$, note that there are $G$ and $H$ s.t. $\bar{d}_{\text{WILT}}(G, H) \neq 0 \wedge \dot{d}_{\text{WILT}}(G, H) = 0$: for example, let $G$ and $H$ be $k$-regular graphs (such as cycles) with different numbers of nodes and identical node and edge attributes. Next, we show $\bar{d}_{\text{WILT}} = d_{\text{WL}}$.

$$\bar{d}_{\text{WILT}}(G, H) = 0 \iff \bar{\nu}^G = \bar{\nu}^H$$
$$\iff \forall \text{ leaf color } c: \quad |\{v \in V_G \mid c_v^{(L)} = c\}| = |\{v \in V_H \mid c_v^{(L)} = c\}|$$
$$\iff \{\!\{c_v^{(L)} \mid v \in V_G\}\!\} = \{\!\{c_v^{(L)} \mid v \in V_H\}\!\}$$
$$\iff d_{\text{WL}}(G, H) = 0.$$

The first equivalence again follows from the fact that weights are positive. $\qquad\square$

Since $d_{\text{MPNN}} \leq d_{\text{WL}}$ holds for any MPNN (Xu et al., 2019), the above theorem implies that $d_{\text{MPNN}} \leq \bar{d}_{\text{WILT}}$ if all edge weights are positive. At first glance, this seems to suggest that $\bar{d}_{\text{WILT}}$ can better align with $d_{\text{MPNN}}$ of any MPNN than $\dot{d}_{\text{WILT}}$ because of its high expressiveness. However, the results in Section 6 and Appendix E show that $\dot{d}_{\text{WILT}}$ is suitable for MPNNs with mean pooling, while $\bar{d}_{\text{WILT}}$ is suitable for MPNNs with sum pooling. Next, we compare $d_{\text{MPNN}}$ and $d_{\text{WILT}}$ in more detail to interpret these results. We start with MPNNs with mean pooling, whose pseudometrics we will call $d_{\text{MPNN}}^{\text{mean}}$.

**Theorem A.10** (Expressiveness of the Pseudometric of MPNN with Mean Pooling). *Suppose $\dot{d}_{WILT}$ and $\bar{d}_{WILT}$ are pseudometrics defined with WILT with some edge weight functions. We assume that all edge weights are positive. Then,*

$$d_{MPNN}^{mean} \leq \dot{d}_{WILT}(< \bar{d}_{WILT}).$$

*Proof.* We first show the left inequality.

$$\dot{d}_{\text{WILT}}(G, H) = 0 \implies \dot{\nu}^G = \dot{\nu}^H$$
$$\implies \forall \text{ leaf color } c: \quad \frac{|\{v \in V_G \mid c_v^{(L)} = c\}|}{|V_G|} = \frac{|\{v \in V_H \mid c_v^{(L)} = c\}|}{|V_H|}$$
$$\implies \forall \text{ leaf color } c: \quad \frac{1}{|V_G|} \sum_{v \in V_G : c_v^{(L)} = c} h_v^{(L)} = \frac{1}{|V_H|} \sum_{v \in V_H : c_v^{(L)} = c} h_v^{(L)}$$
$$\implies \frac{1}{|V_G|} \sum_{v \in V_G} h_v^{(L)} = \frac{1}{|V_H|} \sum_{v \in V_H} h_v^{(L)}$$
$$\implies d_{\text{MPNN}}^{\text{mean}}(G, H) = 0.$$

The first implication follows from the fact that $w(e_{\{c,p(c)\}}) > 0$ for all colors. The third implication follows from Xu et al. (2019) by noting that $c_u^{(L)} = c_v^{(L)} \implies h_u^{(L)} = h_v^{(L)}$ for any MPNN. $\dot{d}_{\text{WILT}} < \bar{d}_{\text{WILT}}$ follows from Theorem A.9. $\qquad\square$

In Section 6 and Appendix E, we show that RMSE($d_{\text{MPNN}}^{\text{mean}}, \dot{d}_{\text{WILT}}$) is smaller than RMSE($d_{\text{MPNN}}^{\text{mean}}, \bar{d}_{\text{WILT}}$). The above theorem and the proof yield an interpretation of the result. In terms of expressiveness, $\dot{d}_{\text{WILT}}$ is a stricter upper bound on $d_{\text{MPNN}}^{\text{mean}}$ than $\bar{d}_{\text{WILT}}$, since the mean pooling and the size normalization are essentially the same procedure. Both ignore the information

about the number of nodes in graphs. When we try to fit $\bar{d}_{\text{WILT}}$ to $d_{\text{MPNN}}^{\text{mean}}$, it is difficult to tune edge parameters so that $\bar{d}_{\text{WILT}}$ can ignore the number of nodes in graphs, but $\dot{d}_{\text{WILT}}$ satisfies this property by definition. This may be the reason why $\dot{d}_{\text{WILT}}$ can be trained to be better aligned with $d_{\text{MPNN}}^{\text{mean}}$ than $\bar{d}_{\text{WILT}}$. A similar discussion can be applied to $d_{\text{WWL}}$ and $d_{\text{WLOA}}$, which are special cases of $\dot{d}_{\text{WILT}}$ and $\bar{d}_{\text{WILT}}$, respectively. Next, we analyze MPNNs with sum pooling.

**Theorem A.11** (Expressiveness of the Pseudometric of MPNN with Sum Pooling). *Suppose $\bar{d}_{WILT}$ is defined with WILT with an edge weight function that assigns a positive value to all edges. Then,*

$$d_{MPNN}^{sum} \leq \bar{d}_{WILT}.$$

*In addition, if $\exists G \in \mathcal{G}$ s.t. $\sum_{v \in V_G} h_v^{(L)} \neq 0$, then*

$$d_{MPNN}^{sum} \lesssim \dot{d}_{WILT}$$

*Proof.* We begin with $d_{\text{MPNN}}^{\text{sum}} \leq \bar{d}_{\text{WILT}}$.

$$
\begin{aligned}
\bar{d}_{\text{WILT}}(G, H) = 0 &\implies \bar{\nu}^G = \bar{\nu}^H \\
&\implies \forall \text{ leaf color } c: \quad |\{v \in V_G \mid c_v^{(L)} = c\}| = |\{v \in V_H \mid c_v^{(L)} = c\}| \\
&\implies \forall \text{ leaf color } c: \quad \sum_{v \in V_G : c_v^{(L)} = c} h_v^{(L)} = \sum_{v \in V_H : c_v^{(L)} = c} h_v^{(L)} \\
&\implies \sum_{v \in V_G} h_v^{(L)} = \sum_{v \in V_H} h_v^{(L)} \\
&\implies d_{\text{MPNN}}^{\text{sum}}(G, H) = 0.
\end{aligned}
$$

Next, we show $d_{\text{MPNN}}^{\text{sum}} \lesssim \dot{d}_{\text{WILT}}$. Let $G$ be a graph that satisfies $\sum_{v \in V_G} h_v^{(L)} \neq 0$. We can consider a graph $H$ that consists of two copies of $G$. Then, $\dot{\nu}^G = \dot{\nu}^H$, since $2\nu^G = \nu^H$ and $2|V_G| = |V_H|$. Therefore, $\dot{d}_{\text{WILT}}(G, H) = 0$. On the other hand,

$$
\begin{aligned}
d_{\text{MPNN}}^{\text{sum}}(G, H) &= \left\| \sum_{v \in V_G} h_v^{(L)} - \sum_{v \in V_H} h_v^{(L)} \right\|_2 \\
&= \left\| \sum_{v \in V_G} h_v^{(L)} - 2 \sum_{v \in V_G} h_v^{(L)} \right\|_2 \\
&= \left\| \sum_{v \in V_G} h_v^{(L)} \right\|_2 \\
&\neq 0.
\end{aligned}
$$

$\square$

In terms of expressiveness, $d_{\text{MPNN}}^{\text{sum}}$ is almost always not bounded by $\dot{d}_{\text{WILT}}$ except for the trivial MPNN which embeds all graphs to zero. In fact, the opposite $\dot{d}_{\text{WILT}} \leq d_{\text{MPNN}}^{\text{sum}}$ holds if the MPNN is sufficiently expressive, e.g. GIN. These analyses may explain why RMSE($d_{\text{MPNN}}^{\text{sum}}, \bar{d}_{\text{WILT}}$) is generally smaller than RMSE($d_{\text{MPNN}}^{\text{sum}}, \dot{d}_{\text{WILT}}$) in Section 6 and Appendix E. No matter how much it is trained, $\dot{d}_{\text{WILT}}$ cannot capture the information about the number of nodes that $d_{\text{MPNN}}^{\text{sum}}$ can. On the other hand, $\bar{d}_{\text{WILT}}$ is expressive enough to capture the information, and thus has a chance of aligning well with $d_{\text{MPNN}}^{\text{sum}}$. Again, a similar reasoning can be applied to $d_{\text{WWL}}$ and $d_{\text{WLOA}}$.

## B. Algorithm to Construct the WILT

Algorithm 2 shows how to build the WILT of a graph dataset $\mathcal{D}$. It starts with the initialization of a root node of the WILT and adds each node color appearing in $\mathcal{D}$ as child of the root. The algorithm then runs $L$ iterations of the Weisfeiler Leman algorithm. Whenever a new vertex color $c_v$ is encountered at some node $v$ in iteration $l$, it is added to the WILT $T_{\mathcal{D}}$ as a child of the color of node $v$ in iteration $l - 1$.

---

**Algorithm 2** Building WILT

---

**Input:** Graph dataset $\mathcal{D}$
**Parameter:** $L \geq 1$
**Output:** WILT $T_\mathcal{D}$
$T_\mathcal{D} \leftarrow$ Initial tree with only the root $r$
**for** $G$ **in** $\mathcal{D}$ **do**
   $c_{\text{pre}} \leftarrow []$                                                   # Keeping colors in the previous iteration
   $c_{\text{now}} \leftarrow []$                                                   # Keeping colors in the current iteration
   /* Add initial colors as children of root */
   **for** $v$ **in** $V_G$ **do**
      **if** $l_{\text{node}}(v) \notin V(T_\mathcal{D})$ **then**
         $V(T_\mathcal{D}) \leftarrow V(T_\mathcal{D}) \cup \{l_{\text{node}}(v)\}$
         $E(T_\mathcal{D}) \leftarrow E(T_\mathcal{D}) \cup \{(r, l_{\text{node}}(v))\}$
      $c_{\text{pre}}[v] \leftarrow l_{\text{node}}(v)$
   /* $L$-iteration WL test on $G$ */
   **for** $l = 1$ **to** $L$ **do**
      **for** $v$ **in** $V_G$ **do**
         $c_v \leftarrow \text{HASH}((c_{\text{pre}}[v], \{\!\{(c_{\text{pre}}[u], l_{\text{edge}}(e_{uv})) \mid u \in \mathcal{N}(v)\}\!\}))$          # Compute iteration $l$ WL color
         /* Add new colors to WILT */
         **if** $c_v \notin V(T_\mathcal{D})$ **then**
            $V(T_\mathcal{D}) \leftarrow V(T_\mathcal{D}) \cup \{c_v\}$
            $E(T_\mathcal{D}) \leftarrow E(T_\mathcal{D}) \cup \{(c_{\text{pre}}[v], c_v)\}$
         $c_{\text{now}}[v] \leftarrow c_v$
      $c_{\text{pre}} \leftarrow c_{\text{now}}$
      $c_{\text{now}} \leftarrow []$
   **return** $T_\mathcal{D}$

---

# C. Experimental Details for Section 4

Here, we present the detailed experimental setup for the results in Figure 2 and Table 1. We conduct experiments on three different datasets: Mutagenicity and ENZYMES (Morris et al., 2020), and Lipophilicity (Wu et al., 2018). We chose these datasets to represent binary classification, multiclass classification, and regression tasks, respectively. For the models, we adopt two popular MPNN architectures: GCN and GIN. For each model architecture, we vary the number of message passing layers $(1, 2, 3, 4)$, the embedding dimensions $(32, 64, 128)$, and the graph pooling methods (mean, sum). This results in a total of $2 \times 4 \times 3 \times 2 = 48$ different MPNNs for each dataset. In each setting, we split the dataset into $\mathcal{D}_{\text{train}}, \mathcal{D}_{\text{eval}}$, and $\mathcal{D}_{\text{test}}$ (8:1:1). We train the model for 100 epochs and record the performance on $\mathcal{D}_{\text{eval}}$ after each epoch. We set the batch size to 32, and use the Adam optimizer with learning rate of $10^{-3}$. $\text{ALI}_k(d_{\text{MPNN}}, d_{\text{func}})$ and the performance metric (accuracy for Mutagenicity and ENZYMES, RMSE for Lipophilicity) are calculated with the model at the epoch that performed best on $\mathcal{D}_{\text{eval}}$. The code to run our experiments is available at `https://github.com/masahiro-negishi/wilt`.

Next, we offer additional experimental results on non-molecular datasets: IMDB-BINARY and COLLAB (obtained from Morris et al., 2020). Figure 7 visualizes the distribution of $\text{ALI}_k(d_{\text{MPNN}}, d_{\text{func}})$ on these datasets and varying $k$. Similar to Figure 2, $\text{ALI}_k$ consistently improves with training. Table 3 also offers results similar to Table 1, showing that there is a positive correlation between $\text{ALI}_k$ of trained MPNNs and their accuracy in general. Figure 8 shows the data used to compute the Spearman's rank correlation coefficient (SRC) in Table 1 and Table 3 for better understanding. Each blue dot

Table 3: Correlation (SRC) between $\text{ALI}_k(d_{\text{MPNN}}, d_{\text{func}})$ and accuracy on $\mathcal{D}_{\text{train}}$ and $\mathcal{D}_{\text{test}}$ under different $k$.

| | IMDB-BINARY | | | | COLLAB | | | |
|---|---|---|---|---|---|---|---|---|
| k | 1 | 5 | 10 | 20 | 1 | 5 | 10 | 20 |
| train | 0.36 | 0.57 | 0.55 | 0.54 | 0.95 | 0.94 | 0.92 | 0.91 |
| test | -0.43 | 0.10 | 0.14 | 0.16 | 0.81 | 0.79 | 0.77 | 0.76 |

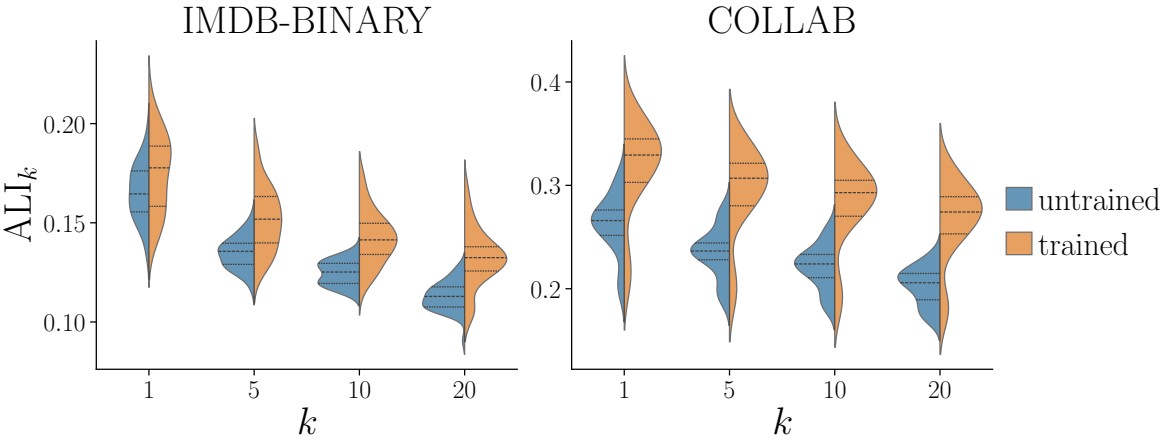

Figure 7: The distribution of $\mathrm{ALI}_k(d_{\mathrm{MPNN}}, d_{\mathrm{func}})$ under different $k$ and datasets.

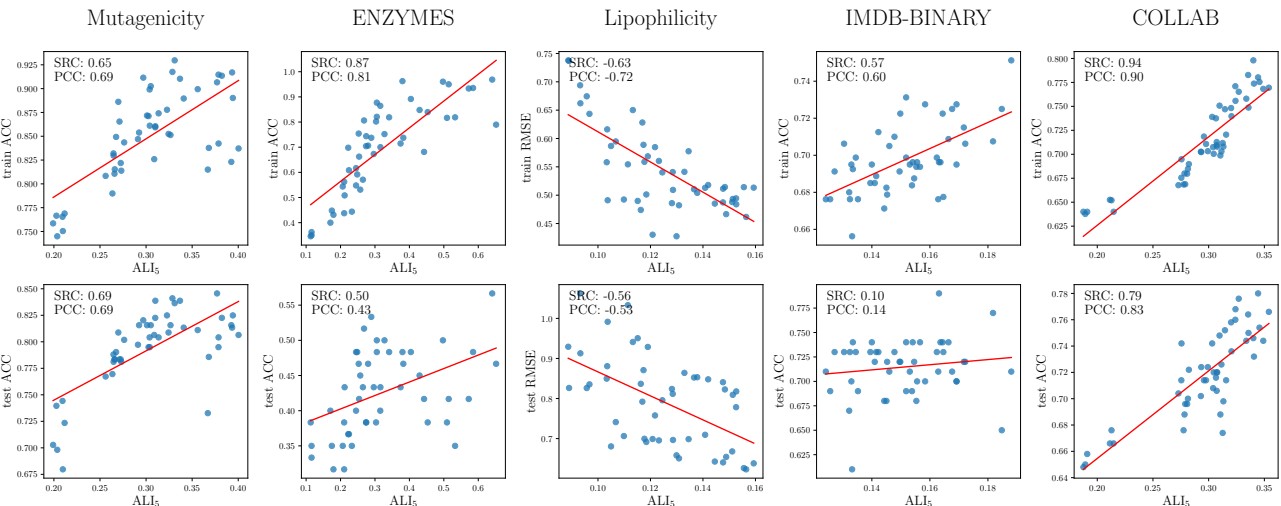

Figure 8: Scatter plots between $\mathrm{ALI}_5(d_{\mathrm{MPNN}}, d_{\mathrm{func}})$ and the performance on the train/test set. SRC and PCC stand for the Spearman's rank correlation coefficient and the Pearson's correlation coefficient, respectively. In general, higher $\mathrm{ALI}_5(d_{\mathrm{MPNN}}, d_{\mathrm{func}})$, i.e., higher alignment between $d_{\mathrm{MPNN}}$ and $d_{\mathrm{func}}$, indicates higher performance.

represents one of the 48 different models. We also fit a linear function and show Pearson's correlation coefficient (PCC). For $\mathrm{ALI}_k$ with $k \neq 5$, similar plots were observed.

## D. MPNN Pseudometric and Structural Pseudometrics

There has been intensive research on graph kernels, which essentially aims to manually design graph pseudometrics $d_{\mathrm{struc}}$ that lead to good prediction performance. Recent studies have theoretically analyzed the relationship between $d_{\mathrm{MPNN}}$ and such $d_{\mathrm{struc}}$, but they only upper-bounded $d_{\mathrm{MPNN}}$ with $d_{\mathrm{struc}}$ (Chuang & Jegelka, 2022), or showed the equivalence for untrained MPNNs on dense graphs (Böker et al., 2024). Therefore, this section examines if $d_{\mathrm{MPNN}}$ really aligns with $d_{\mathrm{struc}}$ in practice, and if the alignment explains the high performance of MPNNs. Specifically, we address the following questions:

**Q1.3** What kind of $d_{\mathrm{struc}}$ is $d_{\mathrm{MPNN}}$ best aligned with?

**Q1.4** Does training MPNN increase the alignment?

**Q1.5** Does a strong alignment between $d_{\mathrm{MPNN}}$ and $d_{\mathrm{struc}}$ indicate high performance of the MPNN?

We first define an evaluation criterion for the alignment between $d_{\text{MPNN}}$ and $d_{\text{struc}}$ to answer them, which is the same as the one used in Section 6 and Appendix E.

**Definition D.1** (Evaluation Criterion for Alignment Between $d_{\text{MPNN}}$ and $d_{\text{struc}}$). Consider a graph dataset denoted by $\mathcal{D}$. Let $\hat{d}_{\text{MPNN}}$ and $\hat{d}_{\text{struc}}$ be normalized versions of $d_{\text{MPNN}}$ and $d_{\text{struc}}$, respectively:

$$\hat{d}_{\text{MPNN}}(G, H) := \frac{d_{\text{MPNN}}(G, H)}{\max\limits_{(G', H') \in \mathcal{D}^2} d_{\text{MPNN}}(G', H')}, \quad \hat{d}_{\text{struc}}(G, H) := \frac{d_{\text{struc}}(G, H)}{\max\limits_{(G', H') \in \mathcal{D}^2} d_{\text{struc}}(G', H')}.$$

We measure the alignment between $d_{\text{MPNN}}$ and $d_{\text{struc}}$ by the RMSE after fitting a linear model with the intercept fixed at zero to the normalized pseudometrics:

$$\text{RMSE}(d_{\text{MPNN}}, d_{\text{struc}}) := \sqrt{\min_{\alpha \in \mathbb{R}} \frac{1}{|\mathcal{D}|^2} \sum_{(G, H) \in \mathcal{D}^2} \left( \hat{d}_{\text{MPNN}}(G, H) - \alpha \cdot \hat{d}_{\text{struc}}(G, H) \right)^2}.$$

The closer the RMSE is to zero, the better the alignment is. Zero RMSE means perfect alignment. That is, $d_{\text{MPNN}}$ is a constant multiple of $d_{\text{struc}}$. Note that the evaluation criterion is different from $\text{ALI}_k$ (Definition 4.1) for measuring the alignment between $d_{\text{MPNN}}$ and $d_{\text{func}}$. There are multiple reasons for this. First, the RMSE is in principle designed for non-binary $d_{\text{struc}}$. Therefore, $\text{RMSE}(d_{\text{MPNN}}, d_{\text{func}})$ is not a meaningful value when $d_{\text{func}}$ is a binary function, which is the case when the task is classification. Second, the computation of $\text{ALI}_k(d_{\text{MPNN}}, d_{\text{struc}})$ is computationally too expensive. We explain this in terms of how many graph pairs we need to compute the distance for. Both the RMSE and $\text{ALI}_k$ require the calculation of the distance between $|\mathcal{D}|^2$ pairs in the original definition. This is too demanding, especially when $d_{\text{struc}}$ is $d_{\text{GED}}$, which is NP-hard to compute. Therefore, in practice, we approximate the RMSE with 1000 randomly selected pairs from $\mathcal{D}^2$. This kind of approximation is difficult for $\text{ALI}_k$. To approximate $\text{ALI}_k$, we first choose a subset $\mathcal{D}_{\text{sub}}$ of $\mathcal{D}$, and then compute $d_{\text{struc}}$ of all pairs in $\mathcal{D}_{\text{sub}}^2$. Even if we set $|\mathcal{D}_{\text{sub}}| = 100$, which is quite small, we still need about 10 times more computation than the RMSE.

We evaluate four structural pseudometrics: graph edit distance ($d_{\text{GED}}$, Sanfeliu & Fu, 1983), tree mover's distance ($d_{\text{TMD}}$, Chuang & Jegelka, 2022), Weisfeiler Leman optimal assignment distance ($d_{\text{WLOA}}$, Kriege et al., 2016), and Wasserstein Weisfeiler Leman graph distance ($d_{\text{WWL}}$, Togninalli et al., 2019). See Appendix A.1 for detailed definitions. $d_{\text{TMD}}$, $d_{\text{WLOA}}$, and $d_{\text{WWL}}$ are pseudometrics on the set of pairwise nonisomorphic graphs $\mathcal{G}$. Only $d_{\text{GED}}$ for strictly positive edit costs is a metric, i.e., $d_{\text{GED}}(G, H) = 0$ if and only if $G$ and $H$ are isomorphic. We will also call $d_{\text{GED}}$ a pseudometric for simplicity. We chose $d_{\text{GED}}$ because it is a popular graph pseudometric. The others were chosen because they are based on the message passing algorithm, like MPNNs, and classifiers based on their corresponding kernels were reported to achieve high accuracy. In addition, $d_{\text{TMD}}$ has been theoretically proven to be an upper bound of $d_{\text{MPNN}}$ (Chuang & Jegelka, 2022). Note that the exact calculation of $d_{\text{GED}}$ is in general NP-hard due to the combinatorial optimization over the set of valid transformation sequences (see Definition A.1). Therefore, in our experiment, we limit the computation time of $d_{\text{GED}}$ of each graph pair $(G, H)$ to a maximum of 30 seconds. If this time limit is exceeded, we consider the lowest total cost at that point to be $d_{\text{GED}}(G, H)$. When we compute the RMSE between $d_{\text{MPNN}}$ and any of $d_{\text{TMD}}$, $d_{\text{WLOA}}$, and $d_{\text{WWL}}$, we set the depth of the computational trees used to compute these $d_{\text{struc}}$ as the number of message passing layers in the MPNN for a fair comparison.

Figure 9 presents the distributions of the RMSE in different datasets (Morris et al., 2020; Wu et al., 2018), $d_{\text{struc}}$, and the readout functions used in MPNN. We followed exactly the same procedure for training and evaluating MPNNs as shown in Appendix C. Each distribution consists of $\text{RMSE}(d_{\text{MPNN}}, d_{\text{struc}})$ of 24 MPNNs with different architectures and hyperparameters. We also provide results for untrained MPNNs to see the effect of training. As can be seen from the plots, the distributions of the untrained and trained MPNNs overlap, and there is no strong and consistent improvement in RMSE after training (answer to **Q1.4**). Regarding **Q1.3**, none of the four $d_{\text{struc}}$ performs best in all cases. The best one depends on the choice of dataset and pooling. One interesting observation is that $d_{\text{MPNN}}$ with sum pooling is more aligned with $d_{\text{WLOA}}$ than $d_{\text{WWL}}$, while the reverse is true for $d_{\text{MPNN}}$ with mean pooling. This difference between pooling methods can be explained by different normalizations of the structural pseudometrics (see Section 5.5 and Appendix A.4).

Another insight from Figure 9 is that the degree of alignment between $d_{\text{MPNN}}$ and $d_{\text{struc}}$ varies by model. To see if the alignment is crucial for the high predictive performance of MPNNs, we examined the SRC between $\text{RMSE}(d_{\text{MPNN}}, d_{\text{struc}})$ of trained models and their performance on the training and test sets. We used accuracy and RMSE as performance criteria. Table 4 shows that the correlation is neither strong nor consistent across settings. Thus the alignment between $d_{\text{MPNN}}$ and $d_{\text{struc}}$ is not a key to high MPNN performance. This answers **Q1.5** negatively.

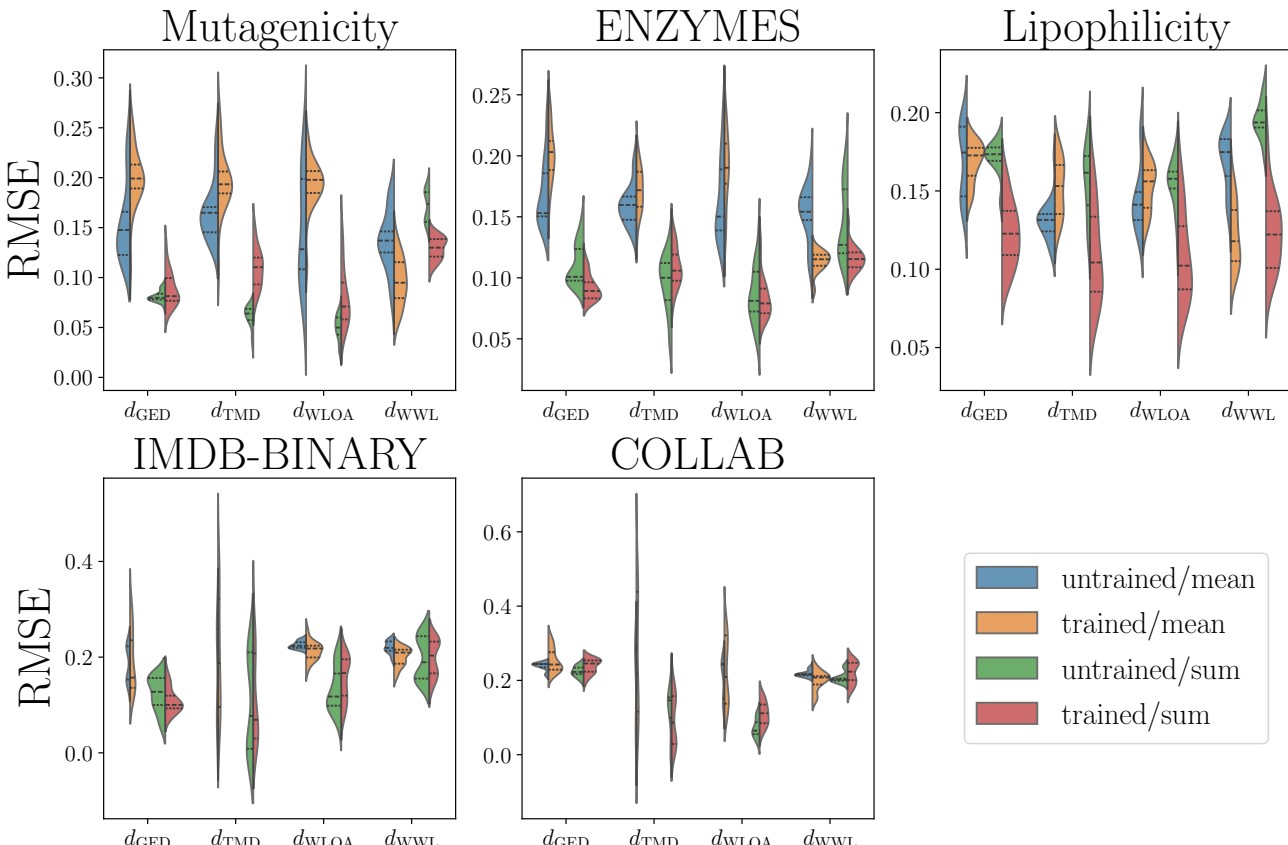

Figure 9: The distributions of RMSE($d_{\mathrm{MPNN}}$, $d_{\mathrm{struc}}$) under different $d_{\mathrm{struc}}$ and datasets. Each color represents whether the MPNNs are trained or not and which graph pooling function they use.

## E. Experimental Details for Section 5

For the experiments in Section 6, we trained 3-layer GCN and GIN with embedding dimensions of 64 on the three datasets. We explored both mean and sum pooling. Each model was trained on the full dataset for 100 epochs using the Adam optimizer with a learning rate of $10^{-3}$. Then, each model was distilled to WILT by minimizing the loss $\mathcal{L}$ defined in Section 5.4. We used the entire data set for $\mathcal{D}$ in $\mathcal{L}$. The distillation was done using gradient descent optimization with the Adam optimizer for 10 epochs. The learning rate and batch size were set to $10^{-2}$ and 256, respectively. See Algorithm 1 for details.

In Table 2, we only show the results for GCN. Here, we show results for GIN in Table 5. The overall trend is the same between Tables 2 and 5: $\dot{d}_{\mathrm{WILT}}$ and $\bar{d}_{\mathrm{WILT}}$ are much better aligned with $d_{\mathrm{MPNN}}$ than $d_{\mathrm{WWL}}$ and $d_{\mathrm{WLOA}}$. In addition, $d_{\mathrm{WWL}}$ and $\dot{d}_{\mathrm{WILT}}$ approximate $d_{\mathrm{MPNN}}$(mean) better, while the opposite is true for $d_{\mathrm{MPNN}}$(sum). We also observed the same trend in the IMDB-BINARY and COLLAB datasets (see Table 6).

Next, we plot the distribution of WILT edge weights after distillation in Figure 10. While the range of edge weights varies by model and dataset, all the distributions are skewed to zero (note that the y-axis is log scale). This suggests that only a small fraction of all WL colors influence $d_{\mathrm{MPNN}}$. In other words, MPNNs build up their embedding space based on a small subset of entire WL colors, regardless of model and dataset.

Finally, we visualize the WL colors with the largest weights, i.e., whose presence or absence influence $d_{\mathrm{WILT}}$ and therefore – by approximation – $d_{\mathrm{MPNN}}$ the most. We use the Mutagenicity dataset as functionally important substructures are known from domain knowledge (Kazius et al., 2005). It should be noted that we only consider colors that appear in at least 1% of all graphs in the dataset. Table 7 and 8 show graphs with substructures corresponding to the WL colors with the top ten largest weights. Table 7 is the result for GCN with sum pooling, while Table 8 is for GCN with mean pooling. If the highlighted subgraph matches one of the seven toxicophore substructures listed in Table 1 of Kazius et al. (2005), we show

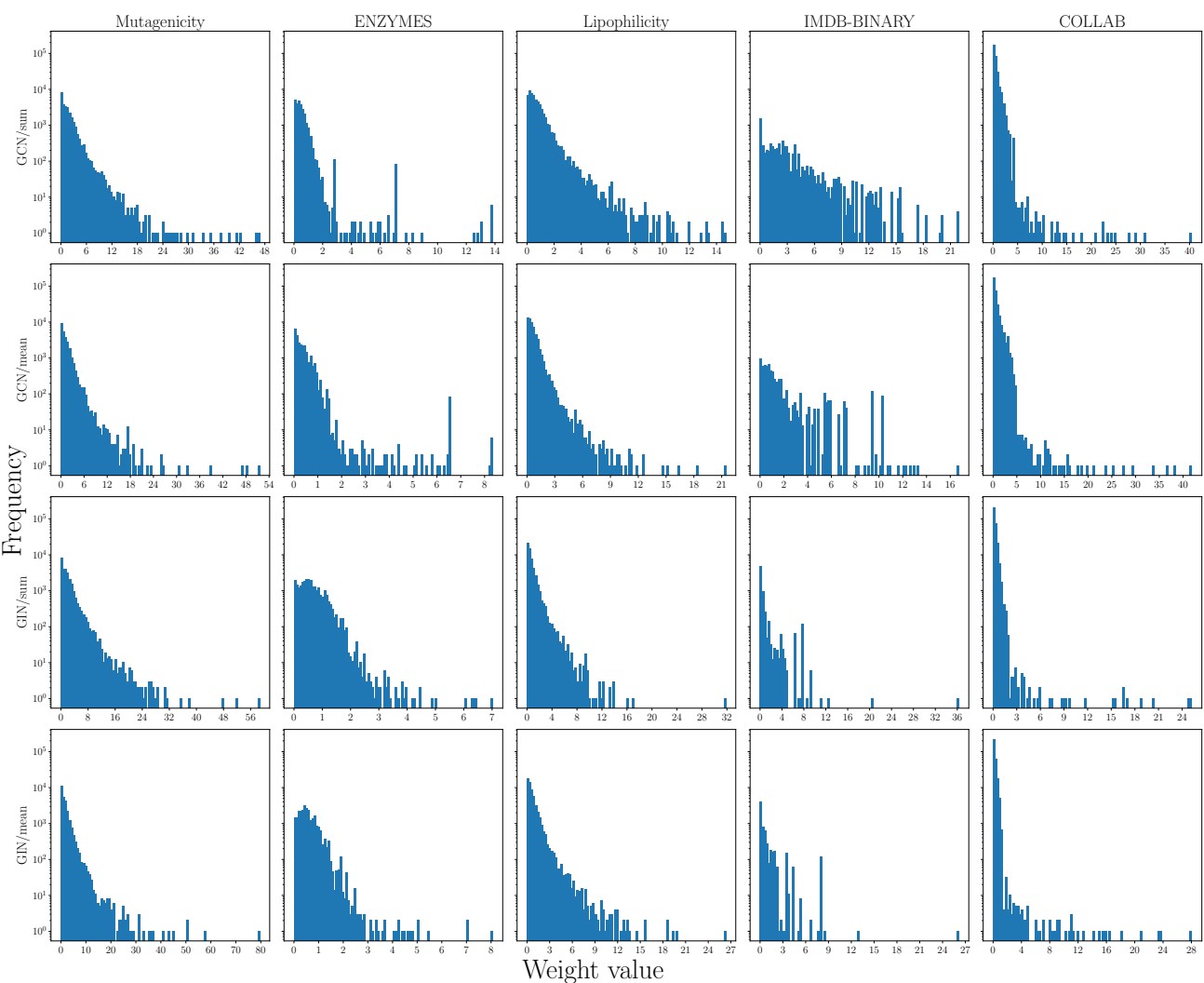

Figure 10: The distribution of edge weights of WILT after distillation from varying models trained on different datasets. The models with sum pooling were distilled into WILT with dummy normalization, while the models with mean pooling were distilled into WILT with size normalization. The log scale y-axis is shared across all plots.

Table 4: Correlation (SRC) between $\mathrm{RMSE}(d_{\mathrm{MPNN}}, d_{\mathrm{struc}})$ and the performance on the training and test sets. Performance was measured based on accuracy for Mutagenicity and ENZYMES, and based on RMSE for Lipophilicity.

| | | Train | | | | Test | | | |
|---|---|---|---|---|---|---|---|---|---|
| | | GED | TMD | WLOA | WWL | GED | TMD | WLOA | WWL |
| Mutagenicity | mean | 0.33 | 0.19 | -0.09 | 0.46 | 0.36 | 0.36 | 0.05 | 0.66 |
| | sum | 0.09 | 0.25 | 0.38 | 0.20 | 0.10 | 0.31 | 0.14 | 0.09 |
| ENZYMES | mean | 0.32 | 0.27 | 0.30 | 0.15 | 0.33 | 0.53 | 0.44 | 0.12 |
| | sum | -0.36 | 0.71 | 0.44 | 0.35 | -0.60 | 0.14 | -0.25 | -0.22 |
| Lipophilicity | mean | -0.57 | -0.61 | -0.58 | -0.46 | -0.55 | -0.77 | -0.65 | -0.65 |
| | sum | -0.12 | -0.62 | -0.54 | -0.32 | -0.53 | -0.85 | -0.85 | -0.64 |
| IMDB-BINARY | mean | 0.04 | 0.10 | -0.53 | -0.20 | -0.30 | -0.31 | -0.39 | 0.41 |
| | sum | 0.37 | 0.65 | -0.54 | -0.51 | 0.01 | 0.22 | -0.23 | 0.02 |
| COLLAB | mean | 0.60 | 0.58 | 0.57 | -0.39 | 0.64 | 0.56 | 0.61 | -0.42 |
| | sum | -0.40 | 0.63 | -0.50 | -0.54 | -0.31 | 0.53 | -0.39 | -0.48 |

Table 5: The mean±std of $\mathrm{RMSE}(d_{\mathrm{MPNN}}, d)$ [$\times 10^{-2}$] over five different seeds. Each row corresponds to GIN with a given graph pooling method, trained on a given dataset.

| | $d_{\mathrm{WWL}}$ | $d_{\mathrm{WLOA}}$ | $\dot{d}_{\mathrm{WILT}}$ | $\bar{d}_{\mathrm{WILT}}$ |
|---|---|---|---|---|
| Mutagenicity | | | | |
| mean | 11.47±0.24 | 17.99±2.79 | 3.70±0.57 | 4.98±0.78 |
| sum | 14.08±0.77 | 13.05±1.44 | 3.86±0.40 | 3.56±0.36 |
| ENZYMES | | | | |
| mean | 11.54±0.30 | 23.71±0.81 | 5.32±0.20 | 7.55±0.24 |
| sum | 12.10±0.84 | 9.94±1.88 | 8.60±0.35 | 3.86±0.68 |
| Lipophilicity | | | | |
| mean | 14.12±0.60 | 16.95±0.52 | 6.31±0.46 | 9.52±0.70 |
| sum | 14.97±0.58 | 13.97±0.75 | 6.49±0.50 | 6.59±0.51 |

the toxicophore name as well. Four out of ten WL colors in Table 7 correspond to toxicophore substructures, while three out of ten in Table 8. These are quite a lot considering that only seven toxicophore substructures are listed in Table 1 of Kazius et al. (2005). Furthermore, there are some colors that not fully but partially match one of the substructures in Kazius et al. (2005). For instance, (6) and (9) in Table 7 and (8) in Table 8 partially match "aromatic nitro", while (7) in Table 8 is part of "polycyclic aromatic system". Note that it is impossible to identify subgraphs that perfectly match these toxicophore substructures, since our method can only identify subgraphs corresponding to a region reachable within fixed steps from a root node. For example, the subgraph in (1) of Table 7 is a region reachable in 2 steps from the oxygen O. This limiation may seem to be a drawback of our proposed method, but in fact it is not. It is natural to identify only subgraphs corresponding WL colors to interpret $d_{\mathrm{MPNN}}$, because MPNNs can only see input graphs as a multiset of WL colors.

Table 6: The mean±std of RMSE($d_{\text{MPNN}}, d$) [$\times 10^{-2}$] over five different seeds. Each row corresponds to a GCN or GIN with a given graph pooling method, trained on IMDB-BINARY or COLLAB dataset.

|  |  | $d_{\text{WWL}}$ | $d_{\text{WLOA}}$ | $\dot{d}_{\text{WILT}}$ | $\bar{d}_{\text{WILT}}$ |
|---|---|---|---|---|---|
| IMDB-BINARY | **GCN** |  |  |  |  |
|  | mean | 16.98±2.06 | 19.04±4.39 | 6.19±1.24 | 7.62±1.27 |
|  | sum | 16.21±2.45 | 12.01±3.81 | 9.08±4.37 | 4.69±3.70 |
|  | **GIN** |  |  |  |  |
|  | mean | 21.32±0.25 | 19.65±0.45 | 2.61±0.34 | 3.09±0.37 |
|  | sum | 23.49±0.42 | 21.23±0.39 | 8.09±0.89 | 0.85±0.13 |
| COLLAB | **GCN** |  |  |  |  |
|  | mean | 16.74±1.83 | 32.60±3.42 | 8.34±1.56 | 19.49±1.61 |
|  | sum | 16.97±0.77 | 6.17±1.26 | 2.21±0.29 | 2.03±0.41 |
|  | **GIN** |  |  |  |  |
|  | mean | 20.37±0.73 | 13.31±0.17 | 3.58±0.65 | 10.54±2.14 |
|  | sum | 25.61±0.42 | 14.38±1.19 | 2.38±1.15 | 1.11±0.21 |

Table 7: Example graphs with highlighted significant subgraphs corresponding to colors with top 10 largest weights. GCN with sum pooling was used. The toxicophore name is shown if the highlighted subgraph matches toxicophore substructures reported in Table 1 of Kazius et al. (2005)

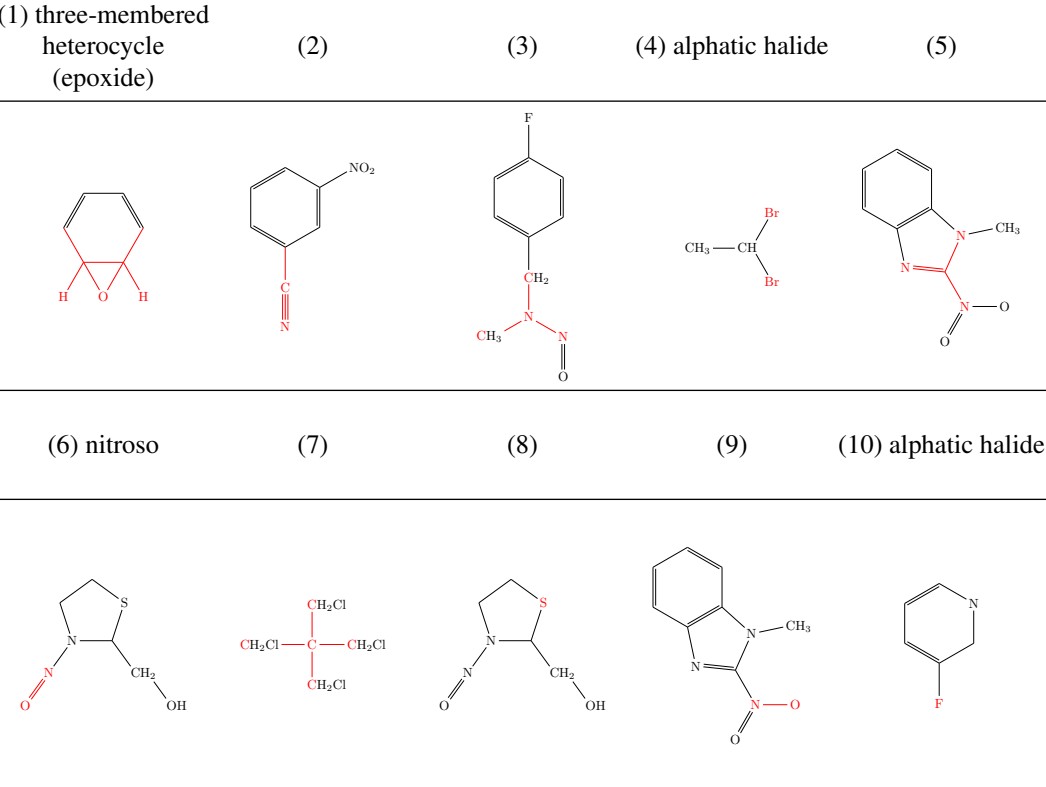

Table 8: Example graphs with highlighted significant subgraphs corresponding to colors with top 10 largest weights. GCN with mean pooling was used. The toxicophore name is shown if the highlighted subgraph matches toxicophore substructures reported in Table 1 of Kazius et al. (2005)

| (1) | (2) three-membered heterocycle (epoxide) | (3) alphatic halide | (4) | (5) |
|---|---|---|---|---|

| (6) | (7) | (8) nitroso | (9) | (10) |
|---|---|---|---|---|

