# OpenReview forum: "WILTing Trees: Interpreting the Distance Between MPNN Embeddings"
_ICML.cc/2025/Conference — ICML 2025 poster_

### Official Review · Reviewer_gzCR · 2025-03-04

**Overall Recommendation:** 4

**Summary:**

This paper investigates how MPNNs learn to embed graphs in a way that captures functional relationships between them. The authors propose a novel interpretable graph distance, the WILTing distance, which effectively approximates the distances between MPNN embeddings. They demonstrate that MPNNs focus on a small subset of WL colors that are functionally important.

**Claims And Evidence:**

Yes

**Essential References Not Discussed:**

No

**Experimental Designs Or Analyses:**

The paper provides comprehensive experiments across multiple datasets and a clear demonstration that the alignment between MPNN distances and functional distances correlates with model performance. However,
1. the authors compare their method only to existing graph kernels and do not compare it against other interpretability methods in GNN, making it difficult to assess the relative advantages of WILT.
2. the experiments do not address how the approach scales to very large graphs or datasets with many graphs, which is crucial for practical applications.

**Methods And Evaluation Criteria:**

Yes

**Other Comments Or Suggestions:**

N/A

**Other Strengths And Weaknesses:**

N/A

**Questions For Authors:**

Q1: I am curious about why the Pearson correlation coefficient is used as a measure of correlation. Is there a linear relationship, or would another method, such as rank correlation, be better for quantifying this relationship?

Q2: I wonder whether this metric is effective for investigating the internal embeddings of MPNNs, rather than just the final embeddings, as previous work[1] (also based on the WL kernel) suggests that consistency in the distance relationship is crucial for MPNN performance.


[1]Liu, et al Exploring Consistency in Graph Representations: from Graph Kernels to Graph Neural Networks. NIPS

**Relation To Broader Scientific Literature:**

No

**Theoretical Claims:**

The paper offers a solid theoretical foundation for understanding MPNN embedding spaces through the lens of metric learning. However, it does not fully explain why some MPNNs achieve better functional alignment than others.

---

> ### Author Rebuttal · Authors · 2025-03-29
>
> Thank you for your review and valuable feedback, which help us clarify our arguments and consider future directions. Below are our answers to your questions and comments.
>
> ---
> > Why some MPNNs achieve better functional alignment than others
>
> Since we focused more on trying out different datasets, we have not conducted in-depth experiments to compare different GNNs. At this point, we share some insights on how hyperparameters affect the functional alignment. We observed that the embedding space of deeper GNNs is more aligned with the functional distance. For example, in Figure 8, deeper models tend to be plotted on the right side of each graph. In future studies, we plan to investigate in more detail what is a key factor contributing to higher functional alignment and thus higher performance. WILT has a nice property that may help in this investigation: The tree structure of WILT is determined only by the dataset and isn't influenced by the GNN. Therefore, different GNNs can be distilled into the same WILT, allowing a direct comparison of the resulting edge weights.
>
> > Comparison against other interpretability methods in GNN
>
> Most interpretation methods aim to find a subgraph that GNN considers important for a prediction for ONE given graph (instance level). On the other hand, our method tries to explain the GLOBAL behavior of GNN by analyzing $d_\mathrm{MPNN}$. The highlighted subgraphs in Figure 6 do not mean important subgraphs for the four specific graphs. They are the ones that strongly influence $d_\mathrm{MPNN}$ in general. This fundamental difference in purpose makes it difficult to compare our method with existing methods. There are also global explanation methods as mentioned in Section 2, but they use decision trees or logical formulas as a language, which are difficult to compare with the subgraphs extracted by our method.
>
> > How the approach scales to very large graphs or datasets with many graphs
>
> We expect WILT to scale to large graphs or datasets, although we have not conducted actual experiments. The number of WILT colors is bounded by the number of nodes in all graphs times the WILT depth. In practice, however, it is much smaller because nodes with the same neighborhood are assigned the same color. As for the computational time, building the WILT and computing the embeddings require linear time in the total number of edges in the dataset. Once the embeddings are prepared as sparse vectors, the computation of $d_\mathrm{WILT}$ is always linear in the number of nodes of the two graphs involved.
>
> Our method is designed for graph-level tasks and cannot be directly applied to node classification, which is a typical task on very large graphs such as social networks. However, we believe that the node embedding space of GNNs can also be distilled to WILT in a similar way, i.e., by tuning the weights to approximate the node embedding distance with a path distance on WILT.
>
> > Q1: Why the Pearson correlation coefficient is used as a measure of correlation
>
> We used Pearson's correlation coefficient (PCC) because it is commonly used and preferable in terms of visualization (see red lines in Figure 8). However, you are right in that there is no valid reason to expect a linear relationship, so we have reanalyzed the results using Spearman's rank correlation coefficient (SRCC). The result is similar to when PCC is used: the functional alignment is more consistently and highly correlated with performance than the structural alignment. Here is the result for the Mutagenicity dataset. We will include these results in the final pdf.
>
> - SRCC between the functional alignment and accuracy (c.f. Table 1)
> || train | test |
> |-|-|-|
> |k=1|0.66|0.70|
> |k=5|0.65|0.69|
> |k=10|0.63|0.68|
> |k=20|0.61|0.67|
> - SRCC between the structural alignment and accuracy (c.f. Table 4)
> ||train/GED|train/TMD|train/WLOA|train/WWL|test/GED|test/TMD|test/WLOA|test/WWL|
> |-|-|-|-|-|-|-|-|-|
> |mean|0.33|0.19|-0.09|0.46|0.36|0.36|0.05|0.66|
> |sum|0.09|0.25|0.38|0.20|0.10|0.31|0.14|0.09|
>
> > Q2: I wonder whether this metric is effective for investigating the internal embeddings of MPNNs, rather than just the final embeddings, as previous work[1] (also based on the WL kernel) suggests that consistency in the distance relationship is crucial for MPNN performance.
>
> In principle, $d_\mathrm{WILT}$ can be trained to approximate the MPNN embedding distance at internal MPNN layers. Our conjecture is that the deeper the layer is, the more the MPNN embedding distance (and thus $d_\mathrm{WILT}$) respects the functional distance. $d_\mathrm{WILT}$ may also keep the consistency proposed in [1], because $d_\mathrm{WILT}$ includes $d_\mathrm{WLOA}$ as a special case, which has been shown theoretically to preserve consistency. However, the proofs provided in [1] require the WILT weights to be constant and hence the results do not immediately extend to our case.
>
> ---
> Please feel free to let us know if you have any other questions or suggestions.

---

> > ### Comment · Reviewer_gzCR · 2025-04-02
> >
> > Thanks for the authors' detailed response. My concerns have been addressed.

---

### Official Review · Reviewer_PD52 · 2025-03-17

**Overall Recommendation:** 3

**Summary:**

The authors of this paper investigate the distance function learned by message-passing neural networks (MPNNs) and introduce a new framework called Weisfeiler Leman Labeling Tree to interpret these distances. Unlike previous work that aligns MPNN embeddings with structural graph distances, the authors focus on task-specific functional distances. Their key contribution is the introduction of the WILT framework, which applies optimal transport techniques to a weighted Weisfeiler-Leman (WL) tree, enabling efficient computation and interpretation of MPNN distances. They demonstrate through experiments in graph classification datasets, that MPNN embeddings capture functionally relevant subgraphs, leading to improved interpretability and performance understanding.

**Claims And Evidence:**

The claims are supported by convincing evidence.

**Essential References Not Discussed:**

The paper provides a detailed  discussion of related work.

**Experimental Designs Or Analyses:**

The experimental design seems valid.

**Methods And Evaluation Criteria:**

The proposed methods and evaluation criteria are suitable for the tasks.

**Other Comments Or Suggestions:**

NA

**Other Strengths And Weaknesses:**

## Strong Points:

- The paper studies the distance metric learned by MPNNs which is crucial for the interoperability of those models.

- The paper moves beyond binary expressivity and addresses how MPNNs learn distance metrics that impact predictive performance.

- WILT can be computed in linear time, making it practical for large-scale graph datasets.

## Weak Points:

-  The paper provides good motivation and theoretical intuition for understanding the learned distance between MPNN embeddings. However, I feel that the experimental results are limited. One of the key limitations is the lack of a systematic ablation study examining how different GNN architectural choices affect the learned embedding distances and the performance of WILTing Trees. For example, how does the number of layers affect the results? Is there a connection with oversmoothing?

- The authors only use GCN and GIN in the experiments. However, many modern GNNs exceed 1-WL expressivity (e.g., higher-order message passing, subgraph-based architectures, additional positional features, graph transformers). It is unclear whether the current findings extend to these more powerful architectures,

**Questions For Authors:**

- Does WILT extend to more expressive GNNs? If so, have you conducted experiments on them?

- Do you see any way to improve GNN performance using insights from WILT, besidesπ the interpretability aspect?

**Relation To Broader Scientific Literature:**

The paper shifts focus from the binary expressiveness of MPNNs (i.e., distinguishing non-isomorphic graphs) to the metric structure of their embeddings. It extends previous theoretical analyses by exploring functional distance alignment (rather than using only structural distance alignment) .

**Theoretical Claims:**

The paper makes several theoretical claims, regarding the properties of the proposed WILT distance, its relationship to existing graph distances, and its alignment with MPNN embeddings. The claims are easy to follow and appear correct.

---

> ### Author Rebuttal · Authors · 2025-03-29
>
> We would like to thank you for your thorough review and valuable feedback. We hope the points below adequately address your concerns.
>
> ---
> > Weakness 1: How different GNN architectural choices affect the learned embedding distances and the performance of WILTing Trees. For example, how does the number of layers affect the results? Is there a connection with over-smoothing?
>
> Since we put more focus on trying out different datasets than on changing GNN architectures, we used only two types of GNNs (GCN and GIN) and poolings (sum and mean). Thus, we admit that the effect of the choice of architecture and hyperparameters is still unclear. At this point, we would like to share some insights on how hyperparameters affect the learned embedding distances. We observed that the deeper the GNN is, the more likely its embedding space is aligned with the functional distance, similar to the observation of [1] Kothapalli et al. mentioned by reviewer n2hL. For example, in Figure 8, deeper models tend to be plotted on the right side of each graph. The connection to over-smoothing is an interesting future work that we did not address during our investigation.
>
> > Weakness 2, Question 1: Does WILT extend to more expressive GNNs? If so, have you conducted experiments on them?
>
> As briefly mentioned in the conclusions, it is possible to construct WILT for higher-order variants of the WL labeling algorithm. We expect that such a higher-order WILT can be used in a similar way to analyze GNNs whose expressiveness is only bounded by higher-order WL variants. However, we have not yet empirically explored this direction.
>
> > Question 2: Do you see any way to improve GNN performance using insights from WILT, besides the interpretability aspect?
>
> This is a very important question. At the moment we don't have an effective way to improve GNN's performance using insights from WILT. However, WILT has a nice property that might help us improve GNN's performance: The tree structure of WILT is determined only by the dataset and isn't influenced by the GNN. Therefore, different GNNs can be distilled into the same WILT. By comparing the resulting edge weights, we may be able to gain insight into why some GNNs perform better than others. This is really a promising future direction.
>
> Although not fully related to your question, we would like to mention the possibility of using WILT to achieve better performance in graph classification or regression. In this study, we train WILT weights by approximating the GNN embedding distance. However, if we find a way to effectively train the weights from scratch, the resulting $d_\mathrm{WILT}$ can be used as a kernel function that is more flexible than other WL-based kernels at no additional computational cost.
>
> ---
> If you have any other questions or suggestions, please let us know and we'll be happy to address them.

---

### Official Review · Reviewer_n2hL · 2025-03-22

**Overall Recommendation:** 3

**Summary:**

The paper investigates (i) the properties of the distances defined for the MPNN based on structural and functional pseudometrics to find the one that explains the high performance of MPNN, and (ii) how MPNNs learn such a structure. The main contribution is the new graph distance based on the Weisfeiler Leman labeling tree (WILT) that is weighted. The nodes in WILT are the colors of the Weisfeiler Leman (WL) test, and the edges connect the parent color with the color that it changes to in the iterations of WL.
The edge weights in WILT are learned such that the graph distance is close to the graph embedding of the message passing neural networks (MPNN). Using this, the authors claim that the edge weights identify the subgraphs that strongly influence the distance between the embeddings of MPNN and use this to interpret the embedding space.

**Claims And Evidence:**

There are two main claims made in the paper based on the analysis: (i) MPNN distance defined on the embedding space is critical to the task performance, and (ii) develops a new distance based on WILT and an algorithm to learn its edge weights to be close to the embedding of MPNN. The edge weights allow to identify the subgraph that yields high performance, thereby providing a way to interpret the embedding space of MPNN.

Claim (i) is expected and intuitive. It is however established clearly using the structural and functional pseudometrics, and evaluation of the alignment to the MPNN embedding on several datasets.

While claim (ii) is evaluated, the computational aspect of the algorithm is not discussed in detail. I understand $d_{WILT}$ is linear in time. But, for a dataset with $n$ graphs with at least $m$ nodes, one needs to apply the WL test algorithm for all the graphs and then build the WILT traversing the resultant graphs from the WL test. What is the complexity of building this tree? and what about its scalability, especially with the number of graphs in the dataset and their sizes?

**Essential References Not Discussed:**

Regarding the first investigation on the properties of the MPNN distances that explain its high performance, [1] studies empirically and theoretically the alignment of the embedding to the task related functional metric (adopting the language from the current paper). This work needs to be discussed.

[1] Kothapalli et al. "A neural collapse perspective on feature evolution in graph neural networks." NeurIPS 2023

**Experimental Designs Or Analyses:**

The experiments are sound and detailed. The plots are especially helpful in conveying the idea and results.

**Methods And Evaluation Criteria:**

The proposed method is novel and the evaluation is sensible. That being said, the purpose of the proposed graph distance is not clear. Is interpretability the primary objective? Then, to strengthen the evaluation, I think the extracted subgraph needs to be compared to the existing methods such as GNNExplainer [1], SubgraphX [2], etc.

[1] Ying et al. GNNExplainer: Generating Explanations for Graph Neural Networks. NeurIPS 2019

[2] Yuan et al. On explainability of graph neural networks via subgraph explorations. ICML 2021.

**Other Comments Or Suggestions:**

'Figure 5' in line 382 second column shouldn't it be Figure 4? Similarly 'Figure 4' in line 411 first column should be Figure 5?

**Other Strengths And Weaknesses:**

**Strengths**

The paper is clearly written, and the claims are supported by experiments.

I found the discussion on the pseudometrics helpful.

The expressivity of WILT is also analyzed which further strengthens the contribution.

**Weaknesses**

The clarity can be improved in some aspects. For instance,

1. The caption of Figure 1 last line is not clear. What is the tuple in the multi-set with $-$ mean?

2. The alignment to structural pseudometric is evaluated in the appendix. It would improve the draft if it is mentioned and referenced in Section 4 as well. Perhaps, the authors can reference Figure 9 and contrast it with Figure 2.

3. Discussion on the limitations of the method is missing.

**Questions For Authors:**

Please check the other sections as well for questions.

One other question: Why do you have $\alpha$ in the RMSE definition? Is it somehow influencing the lower distance for WILT compared to WWL and WLOA?

**Relation To Broader Scientific Literature:**

The key contribution is the WILT based graph distance. The subgraph identified from its edge weights is shown to be functionally important, which is interesting for the interpretability in MPNNs.

**Theoretical Claims:**

The expressivity theorems appear correct. The proofs in the appendix are not checked thoroughly.

---

> ### Author Rebuttal · Authors · 2025-03-29
>
> Thank you for reviewing our paper, including the supplementary materials, and acknowledging our contributions. We address each weakness and question individually below.
>
> ---
> > Complexity of building the tree
>
> Suppose the dataset consists of $|D|$ graphs with at most $|E|$ edges. Then, building WILT with depth $L$ takes $O(|D| \times L \times |E|)$ time. Thus, the complexity is linear in $|D|$ and $|E|$. If the graphs are sparse, i.e. $|V| \propto |E|$, the complexity is also linear in $|V|$. For dense graphs, the complexity is quadratic in $|V|$.
>
> > The extracted subgraph needs to be compared to the existing methods such as GNNExplainer [1], SubgraphX [2], etc.
>
> Most interpretation methods, including [1] and [2], aim to find a subgraph that GNN considers important for a prediction for ONE given graph (instance level). On the other hand, our method tries to explain the GLOBAL behavior of GNN by analyzing $d_\mathrm{MPNN}$. The highlighted subgraphs in Figure 6 do not mean important subgraphs for the four specific graphs. They are the ones that strongly influence $d_\mathrm{MPNN}$ in general. Because of the fundamental difference in purpose, it is difficult to compare our method with [1] and [2]. There are also global explanation methods as mentioned in Section 2, but they use decision trees or logical formulas as a language, which are difficult to compare with the subgraphs extracted by our method.
>
> > Essential References Not Discussed: [1] studies empirically and theoretically the alignment of the embedding to the task related functional metric.
>
> Thank you for bringing this important prior study to our attention. We will add it to the Related Work section in our final version. At this point, we would like to clarify the relationship between our study and [1]. While both investigate the alignment between the embedding distance and the functional distance, there are fundamental differences: Our study examines the graph embedding space of GNNs trained on practical graphs, while [1] analyzes the node embedding space of GNNs trained on graphs generated from a stochastic block model. Nevertheless, both studies reach a similar conclusion: GNNs are trained such that the embeddings respect the functional alignment. It would be interesting to investigate how the degree of alignment between the graph embedding distance and the functional distance changes with layer depth, as [1] did for node embedding.
>
> > Weakness 1: Unclear caption for Figure 1
>
> We apologize for the confusion. $-$ and $\cdots$ represent edges with different labels. We consider a variant of the WL algorithm that takes edge labels into account. This allows WILT to handle graphs with edge labels, such as the molecules in the Mutagenicity dataset. We will clarify the caption in the final pdf.
>
> > Weakness 2: The alignment to structural pseudometric is evaluated in the appendix.
>
> We will include the alignment to $d_\mathrm{struc}$ in the main text in the final pdf.
>
> > Weakness 3: Discussion on the limitations of the method is missing.
>
> One limitation of our study is that we only distilled two GNN architectures (GCN and GIN) with fixed hyperparameters to WILT. Thus, it remains to be seen how different architectures and hyperparameters affect the edge weights of WILT. Another limitation is that our method cannot be directly applied to the analysis of GNNs for node classification on large graphs such as social networks or citation networks. This is because we only deal with graph-level tasks. However, we believe that the node embedding space of GNNs can also be distilled to WILT in a similar way, i.e., by tuning the weights to approximate the node embedding distance with a path distance on WILT. We will include these limitations and possible future work in the final pdf.
>
> > 'Figure 5' in line 382 second column shouldn't it be Figure 4? Similarly 'Figure 4' in line 411 first column should be Figure 5?
>
> We are sorry that these are mistakes. They will be fixed in the final pdf.
>
> > Why do you have $\alpha$ in the RMSE definition?
>
> We use $\mathrm{RMSE}(d_\mathrm{MPNN}, d)$ to measure how well $d$ captures the geometric structure of $d_\mathrm{MPNN}$. We first normalize $d_\mathrm{MPNN}$ so that we can compare or compute the mean of the RMSEs of different MPNNs. The normalization of $d$ is optional as we optimize $\alpha$ anyway. $\alpha$ reflects our belief that the scale of the distance does not affect the importance of the subgraphs. In other words, if $d = \alpha \cdot d_\mathrm{MPNN}$, we consider that $d$ correctly captures $d_\mathrm{MPNN}$. Since $d_\mathrm{WILT}$ is optimized to approximate $d_\mathrm{MPNN}$, $\alpha$ is close to one. However, for $d_\mathrm{WWL}$ and $d_\mathrm{WLOA}$, $\alpha$ is not necessarily close to one. We introduce $\alpha$ for a fair comparison between $d_\mathrm{WILT}$ and $d_\mathrm{WWL}$, $d_\mathrm{WLOA}$.
>
> ---
> Please let us know if you have any further questions or suggestions, and we will be happy to answer them.

---

### Decision · Program_Chairs · 2025-05-01

**Decision:**

Accept (poster)

**Comment:**

The paper proposes an optimal transport based distance between MPNN embeddings. In contrast to existing "structural" distances between embeddings, the proposed distance is task-specific and hence, it is useful to identify relevant subgraphs for a task. All reviewers agree that the theoretical contribution of the work is sound and interesting. Some reviews note that the experiments could be improved, and the author response also shows that there is possibility of further investigation of the proposed WILT framework. Hence, it is an interesting contribution.